# Causal Scene Narration: Inference-Time Text Restructuring for Vision-Language-Action Driving

## Abstract

Vision-Language-Action (VLA) models for autonomous driving receive navigation commands and hazard notices as disconnected text fragments, forcing the model to discover on its own which environmental constraints are relevant to the current maneuver. We introduce **Causal Scene Narration (CSN)**, which restructures VLA text inputs through intent-constraint alignment, quantitative grounding, and structured separation, at inference time with zero GPU cost. [R1-2] We additionally evaluate runtime supervision and training-time preference alignment as experimental conditions rather than as contributions. A multi-town CARLA evaluation (16 routes, 8 towns, $N$=5, [R2-1] 95% bootstrap CIs) shows that CSN improves the [R1-4] CARLA Driving Score (DS, route completion weighted by infraction penalty) by +31.1% on original LMDrive and +24.5% on the preference-aligned variant. [R2-1] A controlled ablation separating information content from text organization shows that information content carries most of CSN's gain on both weight configurations (+6.17 DS on original LMDrive, +6.88 DS on PL-DPO-NLL), with point estimates of 39.1% (95% bootstrap CI [6.6%, 71.4%]) and 13.5% (95% CI [−244%, 41.9%]) of the gain attributable to causal structure; the cross-configuration asymmetry (point difference 25.6%, 95% CI [−23.8%, 306.4%]) is suggestive but not statistically established at $N$=5. A perception noise ablation confirms that CSN's benefit [R3-6] survives distance errors up to $\pm 5\,\text{m}$, speed noise up to $\pm 30\%$, and 20% actor miss rates. [R1-2] We also report a negative result on combining CSN with runtime supervision: a naive direction-conflict monitor never fires on this benchmark and, when paired with CSN, degrades DS via passive control clamping that truncates CSN-guided evasive maneuvers.

## 1 Introduction

Vision-Language-Action (VLA) models combine visual perception with language-conditioned action prediction for end-to-end autonomous driving (Shao et al., 2024; Tian et al., 2024; Mao et al., 2023). In these systems, camera images and LiDAR point clouds are encoded by vision backbones, while natural language provides navigation goals and scene context. Recent results show that text input quality affects driving performance. TLS-Assist (Schmidt et al., 2025) improved LMDrive's driving score by 14.1% simply by injecting structured traffic light messages, and GraphPilot (Schmidt et al., 2026) achieved 15.6% through scene graph serialization, both without any retraining.

A natural interpretation is that the text carries more information. We argue instead that the operative variable is causal structure: whether the text explicitly links what the agent intends to do with what the environment requires it to consider [R1-4] (illustrated in Fig. 1). In causal inference (Pearl, 2009), observing that two variables co-vary does not tell us whether intervening on one will change the other. Similarly, presenting "Turn left" and "Pedestrian ahead" as co-occurring fragments does not tell the model whether the pedestrian is relevant to the turn. Current VLA systems have two related weaknesses.

First, existing systems generate navigation commands and hazard notices as *causally unrelated fragments*. LMDrive (Shao et al., 2024), for example, presents 'Turn left' and 'Pedestrians ahead' separately. The model must independently discover that the pedestrian is relevant *because* the left turn will cross its trajectory. A

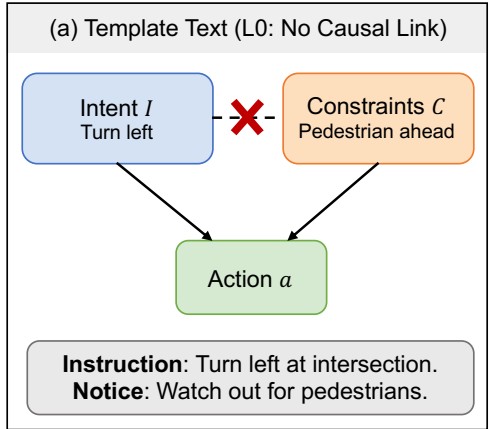 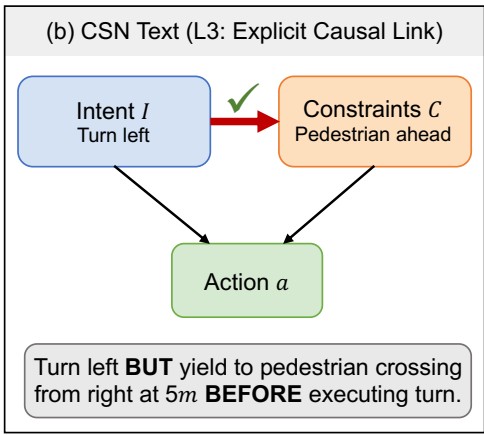

Figure 1: The text structure bottleneck. (a) Template text presents intent $\mathcal{I}$ and constraints $\mathcal{C}$ as independent fragments with no $\mathcal{I} \to \mathcal{C}$ link. (b) CSN restores the causal link via explicit connectives (**BUT**, **BEFORE**).

human instructor would instead say: 'Turn left, *but* yield to the pedestrian at $12\,\mathrm{m}$.' This causally structured utterance links intent to constraint, which is the same structure that DriveVLM (Tian et al., 2024), DriveLM (Sima et al., 2024), and SteerVLA (Gao et al., 2026) each provide through different mechanisms.

Second, preference-aligned models suffer from distribution shift. Preference optimization methods such as DPO (Rafailov et al., 2023) and Multi-PrefDrive (Li et al., 2025) improve in-distribution driving but can overfit to the training environment, hurting generalization to unseen towns. Because inference-time text enrichment leaves model weights unchanged, it sidesteps this failure mode and can complement or even replace training-time alignment.

[R1-2] Our primary contribution targets the first limitation: **Causal Scene Narration** (CSN, §4.2) restructures VLA text inputs around intent-constraint causal alignment, quantitative physical grounding, and structured information separation. The resulting text mirrors the perception-prediction-planning reasoning chain rather than presenting disconnected observations, and the entire pipeline runs on CPU with no additional GPU memory. [R1-2] To probe whether CSN's gains compose with other axes of intervention, we evaluate two additional *experimental conditions*: training-time preference alignment via Plackett–Luce DPO with NLL regularization (PL-DPO-NLL, §4.4), which provides a second weight configuration for ablation; and a runtime direction-conflict monitor (§4.3), which attempts inference-time intervention on top of the VLA output. Neither is presented as a contribution; both are included to test compositional behavior with CSN.

We test this hypothesis through an empirical decomposition (§3.3) showing that when intent and constraints appear as isolated text fragments, the model must discover their relationship through implicit cross-attention, whereas explicitly encoding this relationship via causal connectives lets the model condition on richer structure without any weight change. Our ablation on both weight configurations gives a point estimate of 39.1% [R2-1] (95% bootstrap CI [6.6%, 71.4%], $P(>0)=0.986$) of CSN's improvement on original LMDrive attributable to causal structure, [R2-1] with the lower CI bound just above zero, and 13.5% [R2-1] (95% CI [−244%, 41.9%], $P(>0)=0.664$) on the preference-aligned variant. [R2-1] A bootstrap test for the cross-configuration asymmetry yields a 95% CI [−23.8%, 306.4%] that includes zero, so we present this asymmetry as a suggestive pattern rather than an established effect.

Our contributions are:

1. We identify the absence of intent-constraint links as a limitation of VLA text inputs, organize existing approaches in a taxonomy (L0–L3) by their level of structured linking, and propose CSN, a zero-VRAM text enrichment pipeline built on intent-constraint alignment, quantitative physical grounding, and structured information separation, justified by a controlled ablation.

2. An evaluation across 16 routes, 8 towns, and $N=5$ repetitions with [R2-1] 95% bootstrap confidence intervals (CIs) shows that CSN benefits both tested weight configurations (Original LMDrive and a PL-DPO-NLL variant) with [R2-1] overlapping CIs. [R2-1] Most of the gain on both configurations comes from the added scene information. The share attributable to causal phrasing is measurable on Original LMDrive but not separable from zero on PL-DPO-NLL, and the gap between the two shares is itself within noise (intervals above and in §5.3). At route level the advantage comes from large gains on a few routes (5 wins, 4 ties, 7 losses at a 0.5-DS margin on Original LMDrive) rather than a uniform lift.

3. [R1-1] We re-implement the text-structural patterns of DriveVLM and DriveLM at matched token budget, evaluate them on the same LMDrive benchmark, and find that CSN's natural-language causal connectives transfer to a frozen LMDrive while schema-label and Q&A-graph patterns do not (§5.4).

4. [R1-2] As a negative result on naive runtime supervision, we find that a direction-conflict monitor combined with CSN *degrades* performance via passive control clamping, even though the monitor's specifications never fire. We characterize the mechanism (§5.2.4) and discuss design implications for future safety layers on top of text-enriched VLA models.

## 2 Related Work

### 2.1 VLA Models and Text Structure for Driving

End-to-end autonomous driving increasingly uses VLA models for closed-loop control. GPT-Driver (Mao et al., 2023) reformulated motion planning as language modeling, and showed that autoregressive language models can generate plausible driving trajectories from text-encoded scene states. DriveVLM (Tian et al., 2024) introduced a three-stage Chain-of-Thought (CoT) pipeline consisting of scene description, scene analysis, and hierarchical planning, and showed that when text mirrors the causal reasoning process, long-tail scenario handling improves. DriveLM (Sima et al., 2024) employed graph-structured visual question answering to create logical dependency chains between perception, prediction, and planning nodes, explicitly encoding causal relationships.

LMDrive (Shao et al., 2024) achieves closed-loop control via Q-Former alignment of visual tokens with text, using LLaVA-v1.5 as backbone. Its template-based instruction planner generates navigation commands and hazard notices as causally disconnected fragments, which sits at the minimal end of the text structure spectrum. SteerVLA (Gao et al., 2026) showed that replacing sparse routing commands with fine-grained meta-actions improved driving score by 4.77 points on Bench2Drive, where meta-actions carry implicit causal structure.

Among text-enrichment approaches, those that outperform LMDrive consistently provide text with richer causal structure, whether through CoT decomposition, graph-structured QA, multi-channel separation, or fine-grained meta-actions.

Several studies demonstrate that text enrichment works even without retraining. TLS-Assist (Schmidt et al., 2025) achieved +14.1% DS on LMDrive by injecting structured traffic light messages *without retraining*, showing that LMDrive's pre-trained LLaMA backbone can use structured text never seen during fine-tuning. GraphPilot (Schmidt et al., 2026) achieved +15.6% through scene graph serialization, where relational structure ('pedestrian *is-crossing* ego-lane *conflicts-with* intended left turn') [R2-1] is consistent with the causal structure hypothesis. SimLingo (Renz et al., 2025) found no improvement from post-hoc CoT narration. A possible explanation is that their CoT narrated intended actions without connecting environmental observations to action decisions.

### 2.2 Runtime Safety for Autonomous Driving

Existing runtime safety approaches occupy two categories. *Formal frameworks* include Responsibility-Sensitive Safety (RSS) (Shalev-Shwartz et al., 2017), which defines safe distance envelopes and triggers

proper responses (braking) when violated; Control Barrier Functions (CBFs) (Ames et al., 2019), which enforce forward invariance of safe sets; and Simplex switching (Sha, 2001; Phan et al., 2020), where a verified safety controller runs alongside an unverified high-performance controller. *Runtime verification* methods include STL monitoring (Desai et al., 2017) and shield synthesis (Alshiekh et al., 2018), which check controller behavior against temporal logic specifications.

All these methods operate on physical state (distances, velocities) and are not designed to detect *semantic-level* VLA failures such as direction misinterpretation or hallucinated scene elements. Recent end-to-end driving methods lack explicit runtime safety layers. UniAD (Hu et al., 2023) jointly optimizes perception through planning but errors propagate unchecked. VAD (Jiang et al., 2023) introduces planning constraints that are training-time loss functions not enforced at inference. Chen *et al.* (Chen et al., 2022) and Jaeger *et al.* (Jaeger et al., 2023) document that waypoint predictions fail specifically at junctions due to a "target point shortcut" where models steer toward the nearest GPS waypoint rather than following road geometry. [R1-2] The direction-conflict monitor we evaluate as an experimental condition (§4.3) targets one instance of this failure mode, specifically direction inconsistency during junction approach.

[R1-2] This monitor instantiates a Simplex switch (Sha, 2001) with a safety envelope reformulated for the semantic domain, targeting direction consistency and liveness rather than physical distance maintenance.

### 2.3 Training-Time Safety Alignment

DPO (Rafailov et al., 2023) and its variants optimize policies on preference data but face probability collapse, where chosen action likelihoods decrease during optimization (Pang et al., 2024; Razin et al., 2024). Multi-PrefDrive (Li et al., 2025) applied multi-preference tuning to LLM-based autonomous driving and reported improved in-distribution performance through Plackett-Luce ranking (Plackett, 1975) over multiple candidate actions. NLL regularization provides an explicit likelihood floor against probability collapse. We combine both in our PL-DPO-NLL objective (§4.4).

## 3 Theoretical Foundations

### 3.1 Text Structure as a Performance Bottleneck

[R3-4] We organize text-structural approaches on a causal structure level axis: L0 corresponds to isolated commands with no environmental context; L1 to structured factual information; L2 to entity-level scene graphs or schemas; and L3 to explicit causal dependence between intent and constraints. Section 3.4 details this taxonomy.

In structural causal models (Pearl, 2009), the presence or absence of a directed arrow between two variables encodes a causal assumption. An arrow from $X$ to $Y$ asserts that $X$ influences $Y$, while a missing arrow asserts independence. The same logic applies to VLA text inputs. Let $\mathcal{I}$ denote the navigation intent (*e.g.*, "turn left") and $\mathcal{C} = \{c_1, \dots, c_K\}$ the environmental constraints (*e.g.*, pedestrians, vehicles, traffic lights). The correct driving action $\boldsymbol{a}$ depends not on $\mathcal{I}$ and $\mathcal{C}$ separately, but on their causal interaction, i.e., which constraints are relevant *given* the current intent. A left-turn intent makes a crossing pedestrian safety-critical; the same pedestrian is irrelevant during straight driving.

Template systems present $\mathcal{I}$ and $\mathcal{C}$ as independent text fragments, with no link connecting them. Conceptually, this is equivalent to omitting the edge $\mathcal{I} \rightarrow \mathcal{C}$ from a dependency graph: the text encodes both variables but not their relationship. The LLM must recover the missing dependency internally through multi-layer cross-attention (Vaswani et al., 2017), without any explicit signal indicating which constraints are relevant to the current intent. Prior work supports the broader claim that text acts as a reasoning scaffold. LMDrive's own ablation on the LangAuto benchmark (Shao et al., 2024) showed that adding notice instructions significantly reduces collisions, even though the visual information was always available. TLS-Assist (Schmidt et al., 2025) showed the same pattern: the model could always see traffic lights in the image, but without explicit textual mention, those visual features were insufficiently weighted. These results suggest that text directs the model's attention (Wei et al., 2022) rather than just adding information. Whether *causal*

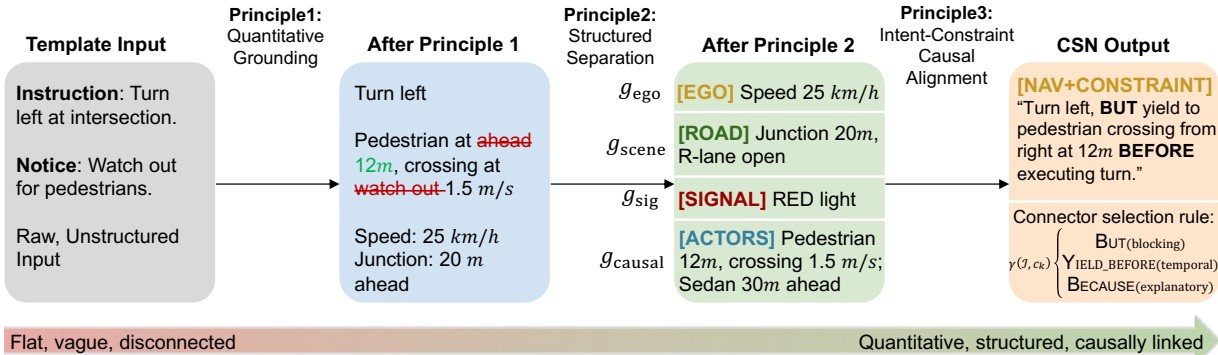

Figure 2: CSN pipeline illustrated on a left-turn scenario. Each principle progressively transforms flat template text into quantitative, structured, and causally linked input.

*structure* within the text provides additional benefit beyond information content is the specific question our CSN vs. Flat Text ablation addresses (§5.2).

CSN restores the missing link by explicitly encoding which constraints are relevant *given* the current intent, using linguistic causal connectives to express this conditional relationship (Fig. 1). With $K$ constraints, the model must evaluate $O(K)$ potential pairings to discover which ones matter for the current intent, whereas CSN pre-selects the $R \ll K$ relevant ones.

### 3.2 How CSN Restructures Text

CSN transforms standard template text through three operations, each targeting a specific weakness of flat VLA text inputs. Fig. 2 illustrates the full pipeline on a left-turn scenario.

**(1) Quantitative physical grounding.** Template terms like 'ahead' and 'watch out' carry no physical dimensions. The LLM cannot tell whether a threat is 10 m or 50 m away, yet this difference dictates the required response (Shalev-Shwartz et al., 2017). CSN replaces all vague qualifiers with exact metric values: distances in meters, speeds in km/h, and timing in seconds. For instance, translating "watch out for pedestrians" into "Pedestrian at 12 m, crossing at 1.5 m/s" allows the model to estimate a 4-second safety window.

**(2) Structured information separation.** Following findings that structured representation outperforms flat descriptions (Tian et al., 2024; Schmidt et al., 2026), CSN organizes the environment state into a four-part sequence (denoted $\langle \cdot \rangle$ for ordered concatenation) mirroring the perception-prediction-planning chain:

$$\boldsymbol{l}_{\text{CSN}} = \big\langle \; g_{\text{ego}}(\boldsymbol{x}), \;\; g_{\text{scene}}(\mathcal{M}, \mathcal{W}),$$
$$g_{\text{sig}}(\mathcal{T}), \;\; g_{\text{causal}}(\mathcal{I}, \mathcal{C}_R, \gamma) \; \big\rangle . \tag{1}$$

Here, $g_{\text{ego}}$ encodes ego-state $\boldsymbol{x} = (v, \omega)$; $g_{\text{scene}}$ describes road topology $\mathcal{M}$ and weather $\mathcal{W}$; and $g_{\text{sig}}$ details traffic signals $\mathcal{T}$. These first three components act as independent observation encoders. The final component, $g_{\text{causal}}$, is where the reasoning occurs: it fuses the navigation intent $\mathcal{I}$ with a filtered set of relevant constraints $\mathcal{C}_R \subseteq \mathcal{C}$, selecting which detections matter for the current maneuver.

**(3) Intent-constraint causal alignment.** The third operation links intent to constraints using explicit causal connectives. Human instructors use connectives to reveal the *nature* of a conflict (*e.g.*, "Turn left, BUT…" vs. "Reduce speed BECAUSE…"). CSN mechanizes this by classifying each relevant constraint $c_k \in \mathcal{C}_R$ into one of three conflict types ($\tau_k$) and assigning a connective via a selection function $\gamma$:

$$\gamma(\mathcal{I}, c_k) = \begin{cases} \text{BUT} & \text{if } \tau_k = \text{blocking} \\ \text{YIELD\_BEFORE} & \text{if } \tau_k = \text{temporal} \\ \text{BECAUSE} & \text{if } \tau_k = \text{explanatory} \end{cases} . \tag{2}$$

To determine $\tau_k$, let $z_k$ denote the spatial zone of the constraint $c_k$, and $\mathcal{Z}_\mathcal{I}$ denote the conflict-side zones for the ego-intent $\mathcal{I}$ (*e.g.*, $\mathcal{Z}_{\text{left-turn}} = \{\text{ahead-left, ahead, left}\}$). The classification follows spatial-temporal rules:

- **Blocking** ($\tau_k = $ blocking): A stationary obstacle occupies the intended path ($z_k \in \mathcal{Z}_\mathcal{I}$). *Example: a stopped vehicle directly ahead.*

- **Temporal** ($\tau_k = $ temporal): A moving actor intersects the intended path ($z_k \in \mathcal{Z}_\mathcal{I}$). The conflict has a time dimension and may resolve if the ego yields. *Example: a crossing pedestrian during a left turn.*

- **Explanatory** ($\tau_k = $ explanatory): The constraint lies outside the immediate conflict zone ($z_k \notin \mathcal{Z}_\mathcal{I}$) but provides necessary context. *Example: a speed limit reduction or a vehicle in an adjacent, non-conflicting lane.*

When a constraint satisfies multiple conditions, priority is assigned as blocking > temporal > explanatory. By embedding these connectives, CSN provides the LLM with explicit causal cues that it is already optimized to process from its natural language pre-training.

[R3-5] The restriction to three conflict types is deliberate. Blocking, temporal, and explanatory conflicts correspond one-to-one to BUT, YIELD_BEFORE, and BECAUSE, connectives that LMDrive's frozen LLaMA backbone already understands from pretraining, and together they cover the four causes of route failure that dominate our 16-route benchmark. A stopped obstacle in the intended path and a red-light stop line are both blocking conflicts rendered with BUT; a crossing or merging actor is a temporal conflict rendered with YIELD_BEFORE; a speed-limit violation is explanatory and rendered with BECAUSE. Nothing in the $\gamma$ family of Eq. (2) is specific to these three connectives, and further ones such as UNLESS or UNTIL would change only the inference-time text renderer, with no retraining of the VLA. Whether three connectives remain sufficient outside driving, for example in indoor manipulation, is untested, and we record this in the limitations.

## 3.3 Empirical Decomposition of Text Utility

CSN's design raises an empirical question: does the model benefit simply from receiving *more* environmental information, or specifically from the *causal organization* of that information? To answer this, we introduce an intermediate baseline $\boldsymbol{l}_{\text{disc}}$ by setting $\gamma = \varnothing$ in Eq. (1), so $g_{\text{causal}}(\mathcal{I}, \mathcal{C}_R, \varnothing)$ provides the same quantitative facts as CSN (distances, speeds, states) but presents them as disconnected fragments without causal connectives. This yields a clean three-way comparison: template $\to$ disconnected $\to$ CSN. The total driving-score gain over the baseline decomposes as:

$$\Delta\text{DS}_{\text{total}} = \text{DS}(\boldsymbol{l}_{\text{CSN}}) - \text{DS}(\boldsymbol{l}_{\text{template}})$$
$$= \underbrace{\left[\text{DS}(\boldsymbol{l}_{\text{disc}}) - \text{DS}(\boldsymbol{l}_{\text{template}})\right]}_{\text{Utility}_{\text{info}}} + \underbrace{\left[\text{DS}(\boldsymbol{l}_{\text{CSN}}) - \text{DS}(\boldsymbol{l}_{\text{disc}})\right]}_{\text{Utility}_{\text{struct}}} . \tag{3}$$

Here, $\text{Utility}_{\text{info}}$ captures the gain from quantitative grounding and structured separation alone, while $\text{Utility}_{\text{struct}}$ isolates the additional gain from explicitly aligning intent with constraints. A positive $\text{Utility}_{\text{struct}}$ shows that VLA performance is bottlenecked by the model's ability to infer causal dependencies from flat text, beyond any limitation from information quantity alone. [R1-4] The intermediate baseline $\boldsymbol{l}_{\text{disc}}$ is the controlled condition that allows us to measure the structural contribution; the empirical estimate is reported in §5.3.

## 3.4 Taxonomy of Text Structure Approaches

We organize existing VLA text approaches by their *causal structure level* (Table 1), defined as follows: L0 provides isolated commands with no environmental context; L1 adds structured factual information; L2 introduces entity-level relationships, scene graphs, or fine-grained action decomposition; and L3 explicitly models the causal dependence between navigation intent and environmental constraints.

Table 1: Taxonomy of VLA text approaches by causal structure level. [R1-5] Top block: each approach's reported gain in its original setting (not directly comparable). Bottom block: apples-to-apples results on our LMDrive benchmark, where the "style" rows apply each method's text-structural pattern to LMDrive at matched token budget (§5.4).

| Approach | Level | VRAM | Retrain | DS Gain | Key Mechanism |
| --- | --- | --- | --- | --- | --- |
| LMDrive template (Shao et al., 2024) | L0 | 0 | – | baseline | Isolated instruction + notice |
| TLS-Assist (Schmidt et al., 2025) | L1 | 0 | No | +14.1% | Structured signal messages |
| GraphPilot (Schmidt et al., 2026) | L2 | 0 | No | +15.6% | Entity-relationship graph text |
| SteerVLA (Gao et al., 2026) | L2 | 0 | Yes | +4.77 pts | Fine-grained meta-actions |
| DriveVLM CoT (Tian et al., 2024) | L3 | High | Yes | – | 3-stage causal chain |
| DriveLM graph QA (Sima et al., 2024) | L3 | High | Yes | – | Graph dependency chains |
| **CSN (ours)** | **L3** | **0** | **No** | **+31.1%** | **Intent-constraint alignment** |
| *Apples-to-apples on the same multi-town LMDrive benchmark (§5.4)* | | | | | |
| DriveVLM-style (compressed) | L3 | 0 | No | −4.7% | Field labels (motion=/influence=) |
| DriveVLM-style (verbatim) | L3 | 0 | No | −74.1% | Verbatim Critical Object Analysis |
| DriveLM-style (compressed) | L3 | 0 | No | −11.4% | Q&A graph with Context prefix |
| DriveLM-style (verbatim) | L3 | 0 | No | −82.7% | Full perception–prediction–planning chain |

All Level 3 approaches share the property of explicitly modeling the conditional dependence between constraints and navigation intent, as formalized in §3.1. [R1-1] Among L3 approaches evaluated on the same LMDrive benchmark (§5.4), CSN is the only one whose causal text format matches LMDrive's training distribution closely enough to provide a positive gain over baseline; the field-label and Q&A-graph patterns inherited from DriveVLM and DriveLM, both of which presuppose a model retrained on those specific schemas, instead degrade LMDrive performance. CSN is, as far as we are aware, the first L3 approach to achieve this intent-constraint alignment entirely at inference time without additional VRAM or model retraining.

## 4 Methodology

### 4.1 System Architecture Overview

[R1-2] Our primary contribution operates at inference time (Fig. 3, Stage 2): Causal Scene Narration (§4.2) restructures text inputs from driving-environment data and runs at zero GPU cost. We additionally evaluate two experimental conditions, treated as ablations rather than contributions. Stage 1 (training time): PL-DPO-NLL (§4.4) fine-tunes the base LLaMA-7B model on preference data, thereby providing a second weight configuration for ablation. Stage 3 (inference time): a runtime direction-conflict monitor (§4.3) tests whether per-frame intervention composes with text-side enrichment.

### 4.2 Causal Scene Narration Pipeline

The CSN pipeline converts driving-environment information into structured natural language following the three principles established in Section 3.2.

#### 4.2.1 Environmental Data Extraction

We extract four categories of environmental data from CARLA's Python API, all computed on CPU: (1) dynamic actor states, including all vehicles and pedestrians within 50 m forward and 15 m lateral, with position and velocity in the ego-vehicle frame, and spatial zone classification (ahead/behind/left/right, near/mid/far); (2) traffic infrastructure, including light state, elapsed timing, and speed limits; (3) road topology, including junction proximity, lane availability, and curvature; and (4) environmental conditions, including precipitation, fog density, wetness, and sun altitude. In this work, we use CARLA's privileged API to isolate and evaluate the impact of text structure independently of perception noise. [R3-3] We discuss replacing this with a vision-based perception stack in §5.5.2.

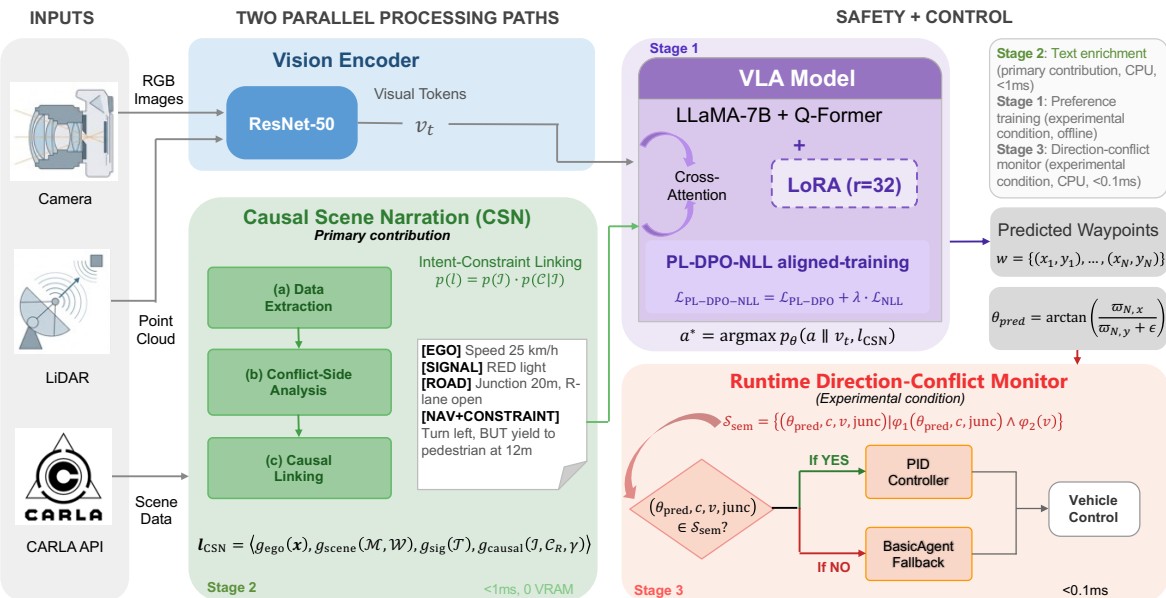

Figure 3: [R1-2] Architecture and pipeline. CSN (Stage 2, our primary contribution) restructures driving-environment information into structured text and runs at zero GPU cost. PL-DPO-NLL training (Stage 1) and the runtime direction-conflict monitor (Stage 3) are evaluated as experimental conditions; see §5.

### 4.2.2 Causal Narration Generation

Given the navigation command $\mathcal{I}$ and detected constraints $\mathcal{C}$, the algorithm first filters $\mathcal{C}$ for relevance to $\mathcal{I}$ via conflict-side analysis, following GraphPilot (Schmidt et al., 2026). Constraints on the conflict side of the intended maneuver (*e.g.*, ahead-left, ahead, and left for a left turn) receive higher priority than those on non-conflict sides. The filtered constraints are then ranked by urgency based on proximity and dynamic state. Finally, each relevant constraint is linked to $\mathcal{I}$ with a causal connective selected by the $\gamma$ function defined in Eq. (2). The entire pipeline runs in $<1\,\mathrm{ms}$ per frame on CPU.

[R3-1] Algorithm 1 formalizes the two-stage filter-then-rank pipeline. The RELEVANCE function combines three signals: a distance-decay term ($\max(0, 50 - d)\,\mathrm{m}$ for vehicles, $\max(0, 40 - d)\,\mathrm{m}$ for pedestrians), a conflict-zone multiplier ($2.5\times$–$3.0\times$ if the constraint is in $\mathcal{Z}_{\mathcal{I}}$, otherwise $0.4\times$–$0.5\times$), and a motion modifier ($2.0\times$ for stopped-ahead vehicles, $1.5\times$ for approaching actors, $3.0\times$ for crossing pedestrians). We keep every constraint whose final priority $r$ exceeds the relevance threshold $\tau = 0$. After ranking by $r$, the renderer greedily packs causal fragments until the per-channel token budget $T_{\max} = 58$ tokens is reached, which corresponds to the bound $K$ on the number of constraints surfaced (variable per frame; typically $K \leq 5$). The token bound, not a fixed $K$, is the binding constraint in practice. Vehicles and pedestrians are also capped at 3 and 2 candidates, respectively, before scoring, which is a defensive pre-filter against pathological CARLA frames.

Table 2 contrasts the three text conditions used in our ablation.

### 4.3 Runtime Direction-Conflict Monitor (Experimental Condition)

[R1-2] We include a runtime monitor as an experimental condition, not as a standalone contribution, to test whether per-frame intervention composes with text-side enrichment. The monitor is structured as a Simplex switch (Sha, 2001), but its empirical effect on this benchmark comes from passive control clamping rather than from any actual envelope violation (§5.2.4); we describe the design here for completeness and discuss the negative result in §5.

---

**Algorithm 1** [R3-1] CSN constraint filtering and ranking

---

**Require:** navigation intent $\mathcal{I}$, detected constraints $\mathcal{C} = \{c_1, \ldots, c_K\}$, conflict-side zones $\mathcal{Z}_{\mathcal{I}}$, relevance threshold $\tau$, token budget $T_{\max}$
**Ensure:** ordered constraint list $\mathcal{C}_R$ rendered into causal narration

 1: $\mathcal{C}_{\text{kept}} \leftarrow \varnothing$
 2: **for** each detected constraint $c \in \mathcal{C}$ (**truncate to $\leq 3$ vehicles, $\leq 2$ pedestrians**) **do**
 3:     $r \leftarrow \text{RELEVANCE}(c, \mathcal{I}, \mathcal{Z}_{\mathcal{I}})$                      ▷ distance-, zone-, and motion-weighted priority
 4:     **if** $r > \tau$ **then**
 5:         $\mathcal{C}_{\text{kept}} \leftarrow \mathcal{C}_{\text{kept}} \cup \{(c, r)\}$
 6:     **end if**
 7: **end for**
 8: Sort $\mathcal{C}_{\text{kept}}$ by $r$ in descending order
 9: $\mathcal{C}_R \leftarrow [\,]$;   $t \leftarrow \text{ESTIMATETOKENS}(\text{intent phrase})$
10: **for** each $(c, r) \in \mathcal{C}_{\text{kept}}$ **do**
11:     $\tau_c \leftarrow$ classify $c$ as blocking, temporal, or explanatory (priority order)
12:     $f \leftarrow \text{RENDER}(c, \gamma(\mathcal{I}, c))$                      ▷ $\gamma$ defined in Eq. (2)
13:     **if** $t + \text{ESTIMATETOKENS}(f) + 2 > T_{\max}$ **then break**
14:     **end if**
15:     Append $f$ to $\mathcal{C}_R$;   $t \leftarrow t + \text{ESTIMATETOKENS}(f) + 1$
16: **end for**
17: **return** $\mathcal{C}_R$

---

Table 2: Text input comparison for a left-turn scenario. Causal connectors shown in bold.

**(a) Template (LMDrive original)**

| | |
|---|---|
| *Instr.* | Turn left at intersection. |
| *Notice* | Watch out for pedestrians. |

**(b) +Flat Text (same facts, no causal links)**

| | |
|---|---|
| *Instr.* | Turn left. Speed 25 km/h. Pedestrian 5m right crossing left. Sedan 12m ahead 30 km/h. RED light. Junction 20m. |

**(c) CSN (causal structure)**

| | |
|---|---|
| *Instr.* | Turn left at intersection, **BUT** yield to pedestrian crossing from right at 5m **BEFORE** executing turn. Maintain distance from sedan ahead. |
| *Notice* | [EGO] 25/30 km/h. [ROAD] Junction 20m, R-lane open. [SIGNAL] RED. [ACTORS] Sedan 12m ahead 30 km/h. Ped 5m R, crossing L. |

### 4.3.1 Simplex Architecture for VLA Driving

The runtime supervisor monitors whether the VLA's output remains inside a semantic safety envelope (Shalev-Shwartz et al., 2017) and triggers a controller switch when violations are detected. We define:

$$\mathcal{S}_{\text{sem}} = \left\{ (\theta_{\text{pred}}, c, v, \texttt{junc}) \mid \varphi_1(\theta_{\text{pred}}, c, \texttt{junc}) \wedge \varphi_2(v) \right\}, \tag{4}$$

where $\varphi_1$ enforces direction consistency with junction-aware gating and $\varphi_2$ enforces liveness, both formalized below. This is realized through a Simplex switching architecture (Sha, 2001) with three components. In the original Simplex terminology, the *High-Performance Controller* (HPC) is the complex but hard-to-verify subsystem, while the *High-Assurance Controller* (HAC) is the simple, conservative fallback. Our instantiation maps these as follows: the **Advanced Controller (AC)**, corresponding to the HPC, is the VLA model (LMDrive + PL-DPO-NLL LoRA), which outputs waypoint trajectories $\boldsymbol{w}_{\text{VLA}} = \{(x_i, y_i)\}_{i=1}^{N}$ in the ego-vehicle frame. The **Baseline Controller (BC)**, corresponding to the HAC, is CARLA's Traffic Manager, a rule-based planner with access to the HD map and ground-truth actor positions that provides a

reliable fallback when the VLA fails. The **Decision Module (DM)** evaluates safety envelope membership $(\theta_{\mathrm{pred}}, c, v, \mathtt{junc}) \in \mathcal{S}_{\mathrm{sem}}$ and switches control authority to the BC when the current state exits the envelope.

The switching logic follows bidirectional Simplex (Phan et al., 2020). Upon exiting $\mathcal{S}_{\mathrm{sem}}$, control transfers from AC to BC for a minimum intervention period $T_{\min}$ of 20 steps, approximately 1 s at 20 FPS. The BC maintains control until the semantic safety envelope is re-entered, at which point authority returns to the AC. Unlike classical safety envelopes that monitor physical distances between vehicles, our DM monitors the semantic consistency between the VLA's predicted actions and the intended navigation command.

### 4.3.2 Safety Specifications

We define two safety properties targeting the dominant VLA failure modes identified by Chen *et al.* (Chen et al., 2022) and Jaeger *et al.* (Jaeger et al., 2023).

**Property $\varphi_1$: Direction consistency (approach phase).** When the route planner issues a turn command $c \in \{\textsc{Left}, \textsc{Right}\}$, the VLA's predicted waypoints must be consistent with the intended direction. Let $\theta_{\mathrm{pred}}$ denote the bearing angle of the last predicted waypoint $\boldsymbol{w}_N$ relative to the ego frame. Since predicted waypoints always lie ahead of the ego vehicle ($w_{N,y} > 0$), arctan suffices without the full atan2 range:

$$\theta_{\mathrm{pred}} = \arctan\left(\frac{w_{N,x}}{w_{N,y} + \epsilon}\right). \tag{5}$$

The direction consistency specification, expressed in Signal Temporal Logic (Desai et al., 2017) where $\square$ denotes "always," requires:

$$\varphi_1 \triangleq \square\Big(\neg\mathtt{in\_junction} \wedge c = \textsc{Left} \implies \theta_{\mathrm{pred}} < -\theta_{\mathrm{thr}}\Big), \tag{6}$$

and symmetrically for $c = \textsc{Right}$. The threshold $\theta_{\mathrm{thr}} = 20°$ separates straight-ahead predictions from turning predictions and was selected empirically based on typical CARLA intersection geometries.

$\varphi_1$ is evaluated only during the *approach phase*, before the vehicle enters the junction. Once inside the junction, the VLA's predicted waypoints naturally flatten in the rotated ego frame because the model correctly predicts 'go forward' relative to its current heading during mid-turn execution. Without junction-aware gating, this flattening triggers false-positive direction conflicts: the monitor infers $\textsc{Straight}$ from flattened waypoints while the route command remains $\textsc{Left}/\textsc{Right}$, causing unnecessary takeovers. Junction boundaries are queried from the CARLA HD map.

**Property $\varphi_2$: Stuck detection.** When throttle is applied, the vehicle must eventually move. Here $\lozenge_{[0,T]}$ denotes "eventually within $T$ steps":

$$\varphi_2 \triangleq \square\Big(\mathtt{throttle} > \tau_{\mathrm{thr}} \implies \lozenge_{[0,\,T_{\mathrm{stuck}}]} \, v > v_{\min}\Big), \tag{7}$$

where $\tau_{\mathrm{thr}} = 0.2$, $v_{\min} = 0.1\,\mathrm{m/s}$, and $T_{\mathrm{stuck}} = 30$ frames ($\approx 1.5\,\mathrm{s}$). A typical stuck situation occurs when the VLA predicts forward motion into a stopped vehicle: the PID controller applies throttle, but the vehicle cannot move.

### 4.3.3 Fallback Policy and Recovery

Upon $\varphi_1$ violation, the DM activates the BC with conservative parameters selected to prioritize safety during intervention: auto lane-change disabled, 5 m following distance, 40% speed reduction. The BC's waypoints $\boldsymbol{w}_{\mathrm{BC}}$ replace the VLA output for $T_{\min}$ steps, after which control returns to the AC. During takeover, steering limits are relaxed to $1.2\times$ the normal maximum to enable trajectory correction, while throttle is capped at $0.6\times$ the normal limit to reduce speed during the intervention.

[R1-3] This architecture is intended as an additional safety layer rather than a formal guarantee: any claim of correctness holds only if the decision module's envelope classification is accurate, and our experiments (§5.2.4) show that $\varphi_1$ and $\varphi_2$ never trigger on this benchmark, so the supervisor's empirical contribution

comes from passive control clamping rather than from intervention itself. The total computational overhead is negligible, under $0.1\,\mathrm{ms}$ per step for map query and angle computation.

[R1-2] We do not introduce an external Simplex baseline because our supervisor itself instantiates the Simplex architecture; demoting the supervisor to an experimental condition makes a separate variant unnecessary.

### 4.4 Training-Time Safety Alignment (Experimental Condition)

We employ PL-DPO-NLL as an *experimental condition* that provides a second weight configuration for ablation, not as a standalone contribution. It combines Plackett-Luce multi-preference ranking (Plackett, 1975) with NLL regularization to address probability collapse during preference optimization (Rafailov et al., 2023; Pang et al., 2024).

#### 4.4.1 Preference Data

We collect 51,124 Plackett-Luce preference samples from CARLA Town01 across 67 route configurations. Each sample contains one expert (chosen) action and 2–3 rejected actions ranked by risk severity. Scene-type distribution: turns (40.2%), normal driving (27.8%), braking scenarios (14.7%), speed adjustment (6.0%), junctions (5.6%), pedestrian interactions (3.1%), and red-light scenarios (2.7%).

#### 4.4.2 Objective Function

The PL-DPO loss generalizes binary DPO to full rankings over $M$ candidates. Let $x = (\boldsymbol{v}_t, \boldsymbol{l})$ denote the multimodal context (visual features and text input):

$$\mathcal{L}_{\text{PL-DPO}} = -\mathbb{E}_{(x,\, y^{(1:M)})} \left[ \sum_{i=1}^{M} \log \frac{\exp(\beta \cdot r_i)}{\sum_{j=i}^{M} \exp(\beta \cdot r_j)} \right], \tag{8}$$

where $r_i = \log \frac{\pi_\theta(y^{(i)}|x)}{\pi_{\text{ref}}(y^{(i)}|x)}$ is the implicit reward, $y^{(1)}$ is the chosen action, and $y^{(2)}, \ldots, y^{(M)}$ are rejected actions ranked by increasing risk. The temperature $\beta$ is scene-adaptive, with higher values for safety-critical scenes to sharpen the preference distribution: $\beta = 0.35$ for turns, pedestrians, and red lights; $\beta = 0.25$ for braking; $\beta = 0.18$–$0.20$ for junctions and speed adjustment; and $\beta = 0.12$ for normal driving. These values were selected via grid search on a held-out validation set from Town01.

The full PL-DPO-NLL objective adds explicit likelihood preservation:

$$\mathcal{L}_{\text{PL-DPO-NLL}} = \mathcal{L}_{\text{PL-DPO}} + \lambda \cdot \mathcal{L}_{\text{NLL}}, \tag{9}$$

where $\mathcal{L}_{\text{NLL}} = -\log \pi_\theta(y^{(1)} \mid x)$ prevents the absolute probability of correct actions from decreasing during preference optimization. We set $\lambda = 0.1$ based on ablation (higher values cause NLL to dominate the preference signal).

#### 4.4.3 Training Configuration

We apply LoRA adapters ($r = 32$, $\alpha = 32$) to all attention and MLP projections (q, k, v, o, gate, down, up) of LLaMA-7B. Training uses AdamW-8bit with learning rate $10^{-5}$, batch size 4 per device with 8 gradient accumulation steps (effective batch 32), 3 epochs, warmup ratio 0.03, and BF16 mixed precision with gradient checkpointing. Training was conducted on $3\times$ NVIDIA RTX 6000 Ada GPUs.

## 5 Experiments

### 5.1 Experimental Setup

Our experiments aim to answer three questions. First, does CSN improve driving performance, and is the improvement robust across different weight configurations? Second, how much of CSN's gain comes from causal structure versus additional information? Third, [R1-2] how do the two non-CSN experimental

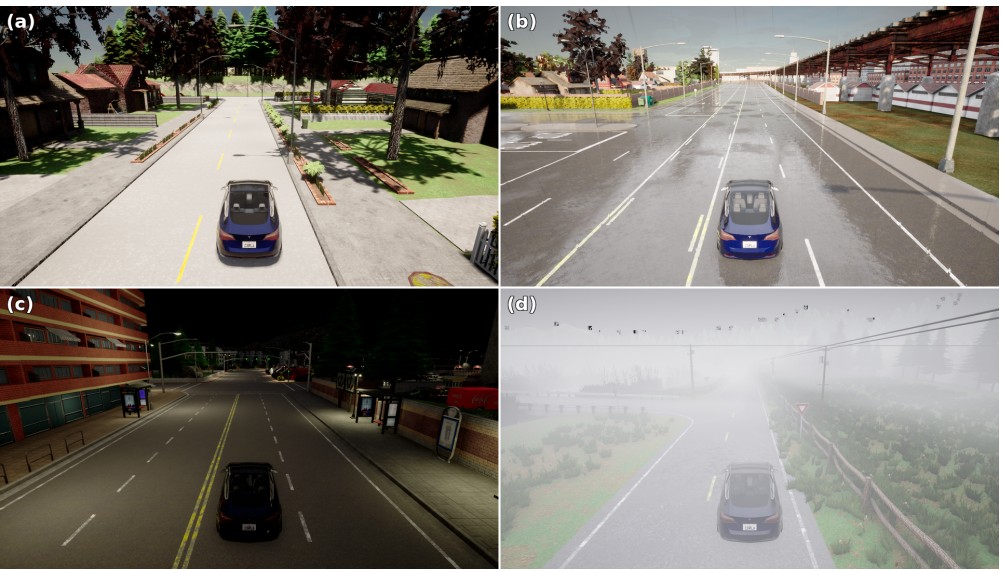

Figure 4: Evaluation environments. (a) Town01, clear day. (b) Town03, heavy rain. (c) Town05, night. (d) Town07, dense fog.

conditions (PL-DPO-NLL preference training and the runtime direction-conflict monitor) compose with CSN, and does either provide additional benefit on top of it?

We evaluate on CARLA 0.9.10 (Dosovitskiy et al., 2017) in closed-loop mode using LMDrive with a LLaMA-7B (Touvron et al., 2023) backbone and ResNet-50 (He et al., 2016) vision encoder, trained with PL-DPO-NLL LoRA on a single NVIDIA RTX 3090 Ti. The benchmark spans 16 routes across 8 towns drawn from the official Leaderboard route set, including 4 night-time routes, 5 rain routes, 3 dense fog routes, and 4 clear daytime routes. This diversity tests generalization across urban layouts, traffic densities, road topologies, and weather conditions absent from single-town benchmarks. Each configuration is evaluated over $N=5$ independent repetitions with distinct random seeds. We report the mean and 95% bootstrap CI; non-overlapping CIs between two configurations suggest a meaningful difference.

We evaluate ten configurations organized hierarchically (Table 3). On the original LMDrive without LoRA, we test (1) original only, (2) +CSN, (3) +Flat Text, and (4) +Semantic Safety. On the PL-DPO-NLL variant with LoRA, we test (5) baseline, (6) +TTC Safety, (7) +Semantic Safety, (8) +CSN, (9) +CSN+Safety, and (10) +Flat Text. Flat Text provides the same factual content as CSN but without causal connectors, enabling the decomposition in Eq. (3) on both weight configurations.

We follow the CARLA Leaderboard metrics. Driving Score (DS) measures route completion weighted by infraction penalty and is the primary metric. Route Completion (RC) measures the percentage of route distance completed. Infraction Score (IS) is a cumulative penalty multiplier where 1.0 means no infractions. Fig. 4 shows representative evaluation environments.

## 5.2 Main Results

Table 3 presents the main results across ten configurations and four dimensions: preference training generalization, CSN robustness, [R1-2] experimental-condition comparison, and component interaction.

### 5.2.1 PL-DPO-NLL Does Not Improve Multi-Town Performance

With $N=5$, PL-DPO-NLL and the original LMDrive have nearly identical mean DS (32.49 vs. 32.54) with heavily overlapping CIs (Table 3), so preference training neither helps nor hurts on aggregate across towns. Since PL-DPO-NLL is trained on 51,124 preference samples collected exclusively from CARLA Town01, the lack of improvement on unseen towns is consistent with the distribution shift documented in DPO literature

Table 3: Multi-town ablation (16 routes, 8 towns, $N$=5). Values are mean $\pm$ 95% bootstrap CI. Best **bold**, second underlined; ties share marking.

| Configuration | DS ($\uparrow$) | RC ($\uparrow$) | IS ($\uparrow$) | $\Delta DS_{orig}$ | $\Delta DS_{base}$ |
|---|---|---|---|---|---|
| LMDrive (original) | 32.54$\pm$3.00 | 48.3$\pm$2.6% | 0.729$\pm$0.034 | — | — |
| + CSN | **42.67$\pm$2.74** | **56.5$\pm$1.7%** | 0.787$\pm$0.028 | +31.1% | — |
| + Flat Text | 38.71$\pm$1.44 | 48.7$\pm$1.9% | **0.828$\pm$0.025** | +18.9% | — |
| + Semantic Safety | 34.10$\pm$2.25 | 45.5$\pm$2.1% | 0.785$\pm$0.038 | +4.8% | — |
| + PL-DPO-NLL | 32.49$\pm$3.34 | 44.9$\pm$2.7% | 0.754$\pm$0.021 | −0.1% | — |
| + TTC Safety | 22.02$\pm$4.07 | 33.7$\pm$3.4% | 0.658$\pm$0.037 | −32.3% | −32.2% |
| + Semantic Safety | 33.17$\pm$1.42 | 44.5$\pm$2.5% | 0.787$\pm$0.022 | +1.9% | +2.1% |
| + CSN | 40.45$\pm$3.79 | 51.9$\pm$4.3% | 0.789$\pm$0.026 | +24.3% | +24.5% |
| + CSN+Safety | 35.74$\pm$1.37 | 48.6$\pm$2.1% | 0.754$\pm$0.020 | +9.8% | +10.0% |
| + Flat Text | 39.38$\pm$2.66 | 49.0$\pm$1.3% | 0.823$\pm$0.047 | +21.0% | +21.2% |

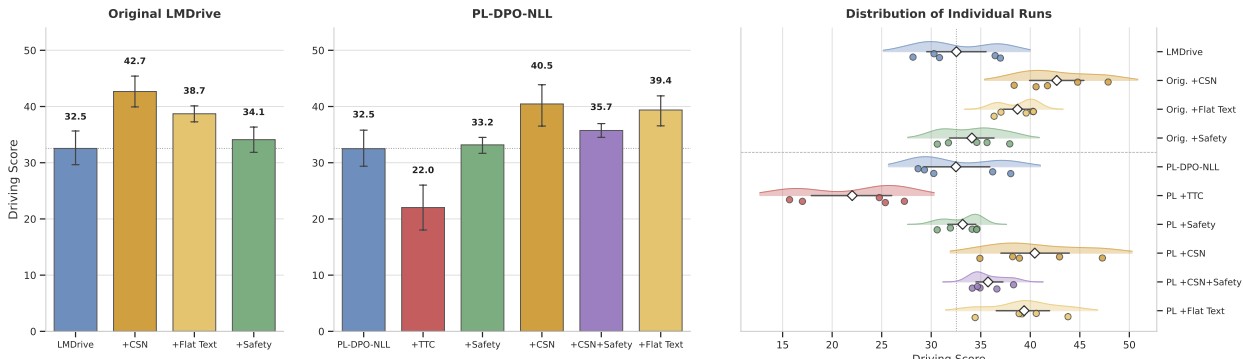

Figure 5: Driving Score comparison across all ten configurations ($N$=5). Left and center: mean DS with 95% bootstrap CI. Right: raincloud plot showing KDE density (shaded), individual runs (dots), mean (diamond), and 95% CI (whisker). Dotted line: original LMDrive baseline.

(Lin et al., 2024), where preference-optimized models overfit to training-distribution patterns at the expense of out-of-distribution generalization. Any gains on Town01 are offset by losses on the remaining seven towns.

PL-DPO-NLL provides here a second weight configuration for the decomposition ablation, not a method we claim improves DS. [R2-1, R2-3] The observed structure-fraction gap (39.1% on Original LMDrive versus 13.5% on PL-DPO-NLL) has a 95% asymmetry CI [−23.8%, 306.4%] that includes zero, so this pattern is consistent with, but does not establish, preference learning partially internalizing causal reasoning. Because CSN operates on the input side without modifying model weights, it does not introduce distribution shift.

### 5.2.2 CSN Benefits Hold Across the Two LoRA Weight Configurations Tested

CSN improves DS by +31.1% on the original LMDrive and +24.5% on PL-DPO-NLL (Table 3). The overlapping 95% CIs between the two CSN-enhanced configurations (42.67$\pm$2.74 vs. 40.45$\pm$3.79) suggest that CSN provides comparable benefits [R3-2] across the two LoRA weight configurations of LMDrive/LLaMA-7B we tested (the original release weights and our PL-DPO-NLL LoRA variant). IS also improves on both variants, which suggests fewer safety violations in novel environments. [R3-2] We do not claim robustness across architectures, model scales, or training recipes beyond these two configurations.

### 5.2.3 Comparing TTC and Direction-Conflict Monitors as Experimental Conditions

The reactive TTC monitor [R1-6] computes the time-to-collision in the standard formulation introduced by Hayward (1972) and triggers emergency braking when this value falls below 2.0 s, a threshold within the range

commonly used in the collision-warning and AEB literature (Vogel, 2003). It achieves the *lowest* DS and IS among all configurations (Table 3). The failure mode is systematic, as frequent false-positive emergency braking causes the vehicle to repeatedly stop and restart, leading to "Agent got blocked" timeout failures. Its wide CIs reflect high variance across repetitions, with performance degrading further as CARLA's stochastic traffic amplifies the over-braking pathology.

[R1-2] The direction-conflict monitor we evaluate as the alternative experimental condition produces narrow CIs and small IS improvements when used in isolation (from 0.729 to 0.785 on original LMDrive; from 0.754 to 0.787 on PL-DPO-NLL, Table 3), since its specifications never fire on this benchmark and it therefore never over-brakes. However, when composed with CSN it degrades DS via passive control clamping (§5.2.4); we therefore do not present it as a method contribution but report the interaction as a negative result.

### 5.2.4 Component Interaction Analysis

CSN and semantic safety are two potentially conflicting safety paradigms. CSN provides *proactive* safety by improving scene understanding so the VLA produces safer decisions, while the semantic supervisor provides *reactive* safety by monitoring outputs and overriding unsafe ones. Their interaction is not additive.

When applied to the unenhanced PL-DPO-NLL baseline, the semantic supervisor has a small positive effect on DS ($+0.68$, Table 3). However, when layered on top of CSN, the same supervisor *degrades* DS by $-4.71$ points ($40.45 \to 35.74$). The tight CI of the CSN+Safety configuration ($\pm1.37$) confirms that the degradation is systematic rather than stochastic.

To diagnose the mechanism, we logged per-frame intervention frequencies across all safety-enabled runs ($N$=3 per config, 16 routes each). The direction-conflict detector ($\varphi_1$) triggered zero interventions across all 96 route-checks in both baseline+Safety and CSN+Safety. Stuck detection ($\varphi_2$) likewise never triggered. The degradation therefore does not arise from explicit safety interventions.

Instead, the cause is passive control clamping. The runtime monitor applies per-frame steering and throttle limits (steer $\leq 0.8$, throttle $\leq 0.9$) on every timestep regardless of whether an intervention is triggered. Without CSN, the VLA produces conservative waypoints with small steering angles, so the clamp rarely binds. With CSN, the VLA receives structured context about upcoming hazards and produces larger anticipatory steering adjustments (early lane changes, preemptive deceleration curves) that exceed the 0.8 steer limit. The clamp truncates these evasive maneuvers, converting them into incomplete corrections that produce worse trajectories than no correction at all. This asymmetric clamping effect explains why the monitor helps the baseline (clamp does not bind) but hurts CSN (clamp truncates beneficial evasive actions). Relaxing or removing the passive clamp when CSN is active would likely resolve this conflict.

### 5.3 Empirical Decomposition of Text Utility

The +Flat Text ablation provides the same factual content as CSN but without causal connectors (BUT, YIELD BEFORE, BECAUSE), enabling an empirical decomposition of the total performance gain into information quantity ($\text{Utility}_{\text{info}}$) and structural organization ($\text{Utility}_{\text{struct}}$) per Eq. (3). Because we run this ablation on both weight configurations, we can test whether the decomposition ratio generalizes.

**Original LMDrive** ($N$=5):

$$\Delta\text{DS}_{\text{total}} = 42.67 - 32.54 = +10.13$$
$$\text{Utility}_{\text{info}} = 38.71 - 32.54 = +6.17 \ (60.9\%)$$
$$\text{Utility}_{\text{struct}} = 42.67 - 38.71 = +3.96 \ (39.1\%).$$

**PL-DPO-NLL variant** ($N$=5):

$$\Delta\text{DS}_{\text{total}} = 40.45 - 32.49 = +7.96$$
$$\text{Utility}_{\text{info}} = 39.38 - 32.49 = +6.88 \ (86.5\%)$$
$$\text{Utility}_{\text{struct}} = 40.45 - 39.38 = +1.08 \ (13.5\%).$$

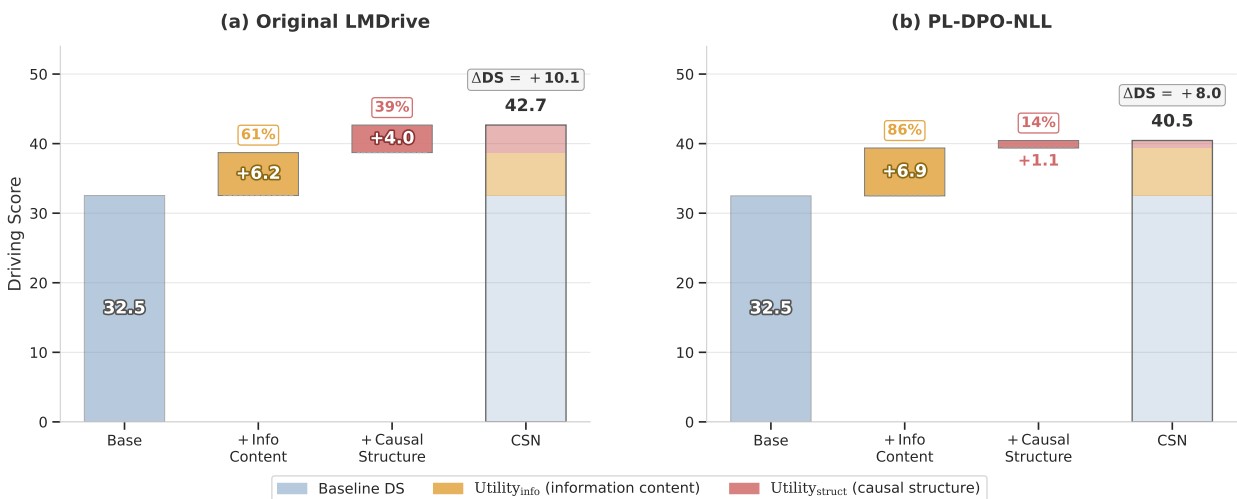

Figure 6: Decomposition of CSN's DS improvement into information content and causal structure contributions on both weight configurations.

[R2-1] On original LMDrive, the structure fraction is 39.1% of the total DS improvement (Fig. 6, 95% bootstrap CI $[6.6\%, 71.4\%]$, $P(>0)=0.986$), with the lower bound just above zero. [R2-1] On PL-DPO-NLL, the point estimate drops to 13.5% with a 95% CI $[-244\%, 41.9\%]$ that crosses zero ($P(>0)=0.664$); the structure fraction on this variant is not statistically distinguishable from zero. [R2-1] Information content contributes comparably in both cases (+6.17 DS vs. +6.88 DS), so most of the gain is information-driven on both configurations. [R2-1] The per-route paired comparison on Original LMDrive yields 5 wins, 4 ties, and 7 losses (mean diff +3.96 DS, 95% CI $[-3.86, 13.41]$), and on PL-DPO-NLL yields 5/6/5 (mean diff +1.08 DS, 95% CI $[-4.44, 6.99]$), indicating that the aggregate mean advantage of CSN over Flat Text arises from a few large per-route wins rather than from uniform improvement across the 16-route suite. [R2-1, R2-3] A bootstrap test for the difference in structure fractions across configurations gives a 95% CI $[-23.8\%, 306.4\%]$ that includes zero, so we report this asymmetry as a suggestive pattern rather than as an established effect at $N=5$; it is consistent with the hypothesis that preference learning on 51k ranked driving samples partially internalizes causal reasoning and reduces the marginal benefit of explicit causal connectors.

On both configurations, +Flat Text outperforms the respective baseline [R2-1] (non-overlapping CIs on both configurations; the margin is narrower on PL-DPO-NLL, 0.9 DS), showing that scene information alone improves performance regardless of weight configuration.

## 5.4 Apples-to-Apples Comparison with DriveVLM and DriveLM Text Patterns

[R1-1] The two L3 approaches surveyed in our taxonomy (Table 1), DriveVLM (Tian et al., 2024) and DriveLM (Sima et al., 2024), both train end-to-end VLMs on their own text formats and report results on different benchmarks (nuScenes, SUP-AD, DriveLM-CARLA). To enable apples-to-apples comparison on our multi-town LMDrive benchmark, we extract their *text-structural patterns* and apply them to LMDrive's text channels while holding the underlying VLA, scene state, route set, and CARLA conditions fixed. We do not retrain DriveVLM or DriveLM end-to-end: doing so would require substantial additional resources and, more importantly, would entangle the contribution of text format with that of training. By keeping the VLA fixed and varying only the text format, we isolate the contribution of text structure from architecture and training.

[R1-1] Each baseline is implemented in two variants. The compressed variant fits LMDrive's 64-token-per-channel budget (64 tokens per channel; 58 remain after reserving the intent phrase, cf. §4.2.2) and matches CSN and Flat Text in token count. The verbatim (faithful) variant relaxes prose to reproduce

Table 4: [R1-1] Apples-to-apples comparison on Original LMDrive (16 routes, 8 towns, $N$=5, 95% bootstrap CIs). All methods share identical scene state and route set; only the text format differs. Compressed variants match CSN's token budget; verbatim variants reproduce paper layouts exactly.

| Configuration | DS ($\uparrow$) | RC ($\uparrow$) | IS ($\uparrow$) | $\Delta$DS |
|---|---|---|---|---|
| LMDrive (original) | 32.54±3.00 | 48.3% | 0.729 | — |
| *DriveVLM-style (3-stage CoT, motion=/influence= fields)* | | | | |
| compressed (matched budget) | 31.02±4.95 | 47.3% | 0.672 | −1.5 |
| verbatim Critical Object Analysis | 8.43±0.79 | 18.4% | 0.472 | −24.1 |
| *DriveLM-style (Graph VQA, $Q_n$ + Context: prefix)* | | | | |
| compressed (matched budget) | 28.82±3.30 | 47.8% | 0.632 | −3.7 |
| verbatim P1→P2→P3 chain | 5.63±0.42 | 13.7% | 0.504 | −26.9 |
| + Flat Text (ours, no causal links) | 38.71±1.44 | 48.7% | 0.828 | +6.2 |
| **+ CSN (ours)** | **42.67±2.74** | **56.5%** | 0.787 | **+10.1** |

the paper's literal layout: DriveVLM's three-line CRITICAL OBJECT / CHARACTERISTIC / INFLUENCE block from Figure 1 of Tian et al. (2024) expanded with $C_s/C_m/C_b$ attribute prose, and DriveLM's full perception–prediction–planning $Q_n$ chain with the (`Context:` ·) prefix from Sima et al. (2024) §3.1. The compressed/verbatim pair tests whether token compression alone explains any performance gap.

[R1-1] Table 4 reports results on Original LMDrive at $N$=5 per configuration. All non-baseline rows share identical scene-state extraction, conflict-side filtering, and constraint ranking. The only experimental variable is the rendered text format.

[R1-1] **Compressed variants match the baseline.** The compressed DriveVLM-style and DriveLM-style configurations reach 31.02±4.95 and 28.82±3.30, respectively, with bootstrap CIs that overlap the LMDrive baseline at 32.54 ± 3.00. Replacing template instructions with DriveVLM's field-style `motion=/influence=` entries or DriveLM's $Q_n$/$A_n$/`Context:` chain neither helps nor hurts. CSN, by contrast, separates from the baseline cluster with a non-overlapping CI (42.67 ± 2.74).

[R1-1] **Verbatim variants degrade performance.** Reproducing the paper layouts in full collapses Driving Score by 24–27 points compared to the baseline, with tight CIs (±0.42 for DriveLM-faithful, ±0.79 for DriveVLM-faithful) that indicate systematic rather than stochastic failure. The verbatim configurations use *more* tokens per frame than CSN (95.1 for DriveLM-faithful vs. 69.4 for CSN), so the gap cannot be attributed to information loss. Inspection of intermediate frames suggests that the schema labels themselves (`Object:`, `Characteristic:`, `Influence:`, $Q_n$ (`Context: A_k`)) are out-of-distribution for LMDrive's frozen LLaMA backbone, which was fine-tuned on plain template instructions and never on schema-prefix text. CSN's connectives (BUT, BEFORE, BECAUSE), in contrast, are pretraining vocabulary that the LLM already knows how to consume.

[R1-1] **Implications for the L3 taxonomy.** Table 1 classifies DriveVLM CoT, DriveLM graph QA, and CSN all at level L3 because each explicitly encodes intent–constraint causal dependence. Our results show that this single label hides a sub-distinction: which surface form encodes the dependence. Schema-field encodings (DriveVLM) and Q&A-graph encodings (DriveLM) work for models trained with those exact schemas but transfer poorly to unmodified LMDrive, while natural-language causal connectives transfer to a frozen LMDrive because they overlap with the pretraining distribution. CSN's contribution is therefore the choice of a transfer-friendly L3 surface form, beyond merely providing L3 structure.

## 5.5 Discussion

### 5.5.1 Why Does Causal Structure Help?

Why does reorganizing the same facts into structured sentences help? Consider a speed reduction scenario. Template text presents 'Reduce speed' and 'Wet road' as unrelated fragments, leaving the LLM to infer whether the wet road is relevant. CSN writes 'Reduce speed BECAUSE wet road reduces braking effectiveness

Table 5: Perception noise ablation on Original LMDrive + CSN. [R2-2] Clean baseline uses $N$=6; each noise condition uses $N$=5. All six conditions share the same 16 routes; the noise RNG seed is fixed at 42 across repetitions and noise levels, and the CARLA traffic-manager seed is at its default. In this table the $\pm$ values are per-rep standard deviations and the bracketed ranges are 95% normal-approximation CIs, whereas Tables 3 and 4 report $\pm$ as the half-width of a 95% bootstrap CI. All four noise-condition CIs overlap with the clean baseline, indicating no statistically significant degradation. [R2-2] The final row is a control in which the same Severe noise is applied to the Flat Text configuration rather than CSN; it shows no noise lift compared to the Flat Text clean mean of 38.71.

| Noise Level | Dist. $\sigma$ | Speed | Miss Rate | $N$ | DS | 95% CI |
|---|---|---|---|---|---|---|
| Clean (privileged) | 0 m | 0% | 0% | 6 | $42.3 \pm 3.5$ | [39.5, 45.0] |
| Mild | $\pm 1$ m | $\pm 10\%$ | 0% | 5 | $45.2 \pm 2.9$ | [42.6, 47.8] |
| Moderate | $\pm 2$ m | $\pm 20\%$ | 0% | 5 | $45.4 \pm 2.7$ | [43.1, 47.6] |
| Severe | $\pm 5$ m | $\pm 20\%$ | 10% | 5 | $46.0 \pm 4.1$ | [42.5, 49.6] |
| Extreme | $\pm 5$ m | $\pm 30\%$ | 20% | 5 | $45.1 \pm 2.7$ | [42.7, 47.4] |
| Flat Text + Severe (control) | $\pm 5$ m | $\pm 20\%$ | 10% | 5 | $37.7 \pm 2.4$ | [35.6, 39.7] |

at current 45 km/h,' where the connective 'BECAUSE' is a direct attention cue. LLaMA has seen millions of such constructions during pre-training on natural text and already knows how to process them. The structured text offloads part of the reasoning from the model's weights to the input.

### 5.5.2 Privileged Information and Deployment Considerations

As noted in §4.2, CSN currently uses privileged simulation data. To test whether the improvement survives under realistic perception errors, we inject calibrated noise into CSN's inputs: Gaussian noise on distance measurements ($\sigma \in \{1, 2, 5\}$ m), multiplicative noise on speed readings ($\pm\{10, 20, 30\}\%$), and random actor miss rates (0–20%). Table 5 shows the results.

[R2-2] All four noise conditions sit above the clean baseline mean (42.3 DS, 95% CI [39.5, 45.0], $N$=6). Under independence and a symmetric null (each condition equally likely above or below clean), the joint event "4-of-4 above clean" has probability $\leq 1/16$. The four noise conditions share the same 16 routes, the noise RNG seed is fixed at 42 across repetitions and noise levels, and the CARLA traffic-manager seed is at its default; under this dependence the true joint probability is somewhat larger than 1/16, but the all-above-clean pattern still merits a mechanism-level explanation rather than dismissal as sampling noise.

[R2-2] We propose three falsifiable mechanism hypotheses, each tied to a predicted outcome under a specific control.

- [R2-2] **(i) Regularization of the text manifold.** Small perturbations of the numeric values rendered into CSN text may push the resulting strings closer to LLaMA's pretraining manifold (a 12 m measurement is more pretraining-frequent than the original 11.83 m). *Falsifier:* the all-above-clean pattern should vanish when the noise is applied only to the textual rendering layer, leaving the underlying perception measurements unchanged.

- [R2-2] **(ii) Asymmetric-cost buffer.** Even with zero-mean Gaussian distance noise (clipped at a 1 m floor), overestimating the gap to an actor is operationally cheaper than underestimating it: CSN encodes any small remaining gap as an earlier YIELD_BEFORE, so positive distance perturbations cost little while negative ones cost more. *Falsifier:* the effect should reverse (degrade below clean) under negative-only noise injection that systematically shortens the reported distance.

- [R2-2] **(iii) Degenerate-tie breaking.** Jitter in the numeric tokens perturbs LLM decoding away from local plateaus, so tied or near-tied candidates are broken in slightly different ways across repetitions. *Falsifier:* because pure baseline LMDrive receives no dynamic numeric text for the noise to act on, the appropriate control is Flat Text with the same noise injected; any text-generic tie-breaking lift should reproduce there.

[R2-2] There is no monotone trend in noise magnitude. Severe (46.0) and Extreme (45.1) differ by less than the bootstrap noise, so we do not claim a saturating fit. The noise sweep alone neither rules out nor confirms any of (i)–(iii).

[R2-2] Because (i) and (iii) are text-generic while (ii) requires CSN's causal rendering, the Flat Text control named in (iii) discriminates between them. Under the Severe setting ($N=5$, identical noise parameters and RNG seed; final row of Table 5), Flat Text scores $37.66 \pm 2.36$ versus $38.71 \pm 1.88$ clean ($\pm$ is the rep-level SD; the $38.71 \pm 1.44$ in Table 3 is a 95% bootstrap CI half-width). The difference of $-1.05$ DS (paired bootstrap 95% CI $[-3.35, +1.38]$, $P(\text{diff} > 0)=0.19$) shows no lift, in contrast to the $+3.7$ DS lift under CSN ($42.3 \rightarrow 46.0$). Given the wide $N=5$ intervals and the shared noise seed, we read this as disfavoring (i) and (iii) rather than confirming (ii).

[R2-2] Confirming (ii) directly would require sign-controlled noise and $N \geq 20$ runs per condition. We attempted $N=10$, but compute-time constraints within the discussion window limited us to $N=6$ for the clean condition and $N=5$ elsewhere, so we leave both to future work.

### 5.5.3 Why LMDrive as the Evaluation Platform

Our experiments use a single VLA architecture (LMDrive with LLaMA-7B). LMDrive is currently the only open-source VLA that satisfies three requirements simultaneously: (1) it accepts free-form text input that CSN can modify, (2) it supports closed-loop CARLA evaluation with the standard Leaderboard protocol, and (3) its training pipeline is publicly available, enabling the PL-DPO-NLL ablation condition. Other VLA models such as DriveVLM (Tian et al., 2024) and Bench2Drive (Jia et al., 2024) either lack open-source training code, do not accept free-form text, or use proprietary evaluation setups that prevent controlled comparison. CSN's mechanism (restructuring text with causal connectors that LLMs already understand from pretraining) is architecture-agnostic in principle, but validating this claim requires additional open-source VLA platforms with closed-loop evaluation capabilities.

### 5.5.4 Robustness Across Weight Configurations

As established in §5.2, the overlapping CIs between the original and preference-aligned CSN configurations suggest that CSN's overall benefit [R3-2] does not depend on either of the two LoRA weight configurations tested. However, the empirical decomposition reveals an asymmetry: causal structure accounts for 39.1% [R2-1] (95% CI [6.6%, 71.4%]) of CSN's gain on original LMDrive but only 13.5% [R2-1] (95% CI [$-244\%$, $+41.9\%$]) on PL-DPO-NLL. [R2-1] The direct bootstrap on the difference between the two structure fractions ($+25.6\%$, 95% CI $[-23.8\%, +306.4\%]$) includes zero. We note that on PL-DPO-NLL, the CSN vs. Flat Text CIs overlap substantially ($40.45\pm3.79$ vs. $39.38\pm2.66$), so the 13.5% figure is not statistically distinguishable from zero. On original LMDrive the overlap is marginal ($42.67\pm2.74$ vs. $38.71\pm1.44$), supporting a meaningful structural contribution in that setting. [R2-3] This pattern is consistent with — but does not establish — preference learning partially internalizing the causal reasoning that explicit text structure otherwise provides. CSN should be compatible with other LMDrive-family checkpoints without further weight updates, though the balance between information and structural contributions may vary. We have not tested whether this transfers to other architectures such as DriveVLM.

### 5.5.5 Benchmark Difficulty and Weather Effects

The benchmark includes 4 night routes, 5 rain routes, and 3 dense fog routes alongside clear daytime conditions. Baseline DS values (32.54 for original LMDrive, 32.49 for PL-DPO-NLL) are well below the 50–60 DS typical of single-town clear-weather evaluations. CSN's $+31.1\%$ improvement holds under these harder conditions.

## 6 Conclusion

We introduced Causal Scene Narration (CSN), a framework that restructures VLA text inputs around intent-constraint causal alignment at zero GPU cost. Through a multi-town CARLA evaluation (16 routes across 8

towns, $N=5$ independent repetitions [R2-2] ($N=6$ for the clean noise-ablation condition) with 95% bootstrap confidence intervals), we establish four main findings.

First, CSN improves DS on both the original LMDrive ($+31.1\%$) and the preference-aligned variant ($+24.5\%$), with overlapping CIs consistent with benefits robust across the [R3-2] two LoRA weight configurations of LMDrive/LLaMA-7B we tested. Second, a controlled ablation on both configurations shows that [R2-1] information content carries most of CSN's gain on both weight configurations ($+6.17$ DS on Original LMDrive, $+6.88$ DS on PL-DPO-NLL); the structure fraction is 39.1% [R2-1] (95% bootstrap CI [$6.6\%, 71.4\%$]) on Original LMDrive and 13.5% [R2-1] (95% CI [$-244\%, 41.9\%$]) on PL-DPO-NLL[R2-1, R2-3], and a bootstrap test for the cross-configuration asymmetry yields a 95% CI [$-23.8\%, 306.4\%$] that includes zero. The asymmetry is therefore consistent with, but does not establish, preference learning partially internalizing causal reasoning. [R1-1] Third, an apples-to-apples comparison with DriveVLM-style and DriveLM-style text-structural patterns (§5.4) shows that CSN's natural-language causal connectives transfer to a frozen LMDrive while schema-label and Q&A-graph patterns do not; this isolates the contribution of CSN's specific surface form within the L3 taxonomy class.

Fourth, a perception noise ablation shows that CSN's benefit is robust to distance errors up to $\pm 5\,\mathrm{m}$, speed noise up to $\pm 30\%$, and 20% actor miss rates, indicating that the improvement [R2-2] does not depend on numerically precise perception inputs.

[R1-2] We additionally report a negative result on layering a runtime direction-conflict monitor on top of CSN. The monitor's specifications never fire on this benchmark. Its empirical effect comes from passive control clamping, which truncates CSN-guided evasive steering and degrades DS rather than improving it. Relaxing the clamp when CSN is active would likely resolve this conflict, and the same caution applies to any runtime supervisor layered on top of a text-enriched VLA.

**Limitations.** (1) Our evaluation is simulation-based; while noise injection experiments (§5.5.2) show CSN is robust to perception errors, real-world deployment requires integration with an actual perception pipeline[R3-6], and a residual sim-to-real gap remains because all CSN inputs are derived from CARLA's privileged API. The noise ablation is the only sim-to-real evidence we offer, and it does not substitute for on-vehicle validation. (2) Experiments use a single model architecture and scale (LMDrive with LLaMA-7B)[R3-2], and we have tested only two LoRA weight configurations of this architecture. (3) [R1-2] The runtime supervisor we evaluate as an experimental condition uses fixed control limits that do not adapt to CSN-enhanced context quality, which is itself the subject of the negative result above. [R3-5] (4) The $\gamma$ connective set is restricted to three causal connectives (BUT, YIELD_BEFORE, BECAUSE), which is sufficient for the conflict structure of this driving benchmark but is untested for non-driving scene types. [R2-2] (5) Compute-time constraints within the discussion window limited the clean perception-noise replication to $N=6$ and noise conditions to $N=5$. We attempted $N = 10$; $N \geq 20$ would be required to resolve the all-above-clean pattern with adequate statistical power.

**Future Work.** (1) Integrating CSN with a vision-based perception pipeline for real-world validation. (2) Extending CSN to other VLA architectures and model scales. (3) [R1-2] If runtime supervision is layered on top of text-enriched VLAs, the passive control clamps must be made CSN-aware to avoid the truncation effect documented in §5.2.4. (4) Extending the empirical decomposition to token-level VLA architectures, where action distributions are directly measurable, enabling a strict information-theoretic validation of $\text{Utility}_{\text{struct}}$.

**Ethics Statement.** CSN aims to improve VLA driving through better text conditioning. The system is evaluated exclusively in simulation; real-world deployment would require extensive additional testing. All experiments use the open CARLA simulator with no human subjects involved.

**Reproducibility Statement.** All experiments use the publicly available CARLA 0.9.10 simulator and the open-source LMDrive codebase. Evaluation follows the standard CARLA Leaderboard protocol with 16 routes across 8 towns. Each configuration is run $N=5$ times [R2-2] ($N=6$ for the clean noise-ablation condition) with fixed random seeds; we report bootstrap 95% CIs throughout. The CSN text generation

pipeline, safety supervisor implementation, evaluation scripts, and all trained LoRA weights will be released upon acceptance.

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
