# OpenReview forum: "Causal Scene Narration with Runtime Safety Supervision for Vision-Language-Action Driving"
_TMLR — Under review for TMLR_

### Review · Reviewer_uZ8t · 2026-04-24

**Summary Of Contributions:**

The paper proposes Causal Scene Narration (CSN), an inference-time text restructuring pipeline for VLA driving models. CSN transforms flat template text inputs into structured natural language organized around three principles:

1. quantitative physical grounding

2. structured information separation into ego/scene/signal/causal components

3. intent-constraint alignment via explicit causal connectives (BUT, YIELD_BEFORE, BECAUSE)

The pipeline runs on CPU at <1 ms per frame with zero VRAM overhead.

Evaluation on CARLA (16 routes, 8 towns, N=5, 95% bootstrap CIs) shows CSN improves Driving Score by +31.1% on original LMDrive and +24.5% on a preference-tuned variant.

The creation of the CSN is well motivated, and experiments establish the benefits from causality clearly and rigorously. The benefits and the problem statement regarding CSN are useful and relevant to the TMLR community. The paper is mostly well written.

The authors also introduce the multi-town evaluation which is a ten-config ablation that relies on 16 routes, 8 towns. This in itself would be useful to the community.

**Audience:**

Yes

**Audience Explanation:**

Majorly yes.

**Claims And Evidence:**

No

**Claims Explanation:**

Things that need to be improved before the paper is up to TMLR standard:

### **Recent Baselines not compared against**

Section 3.4 discusses recent works and highlights that CSN uses the highest level of causal linking. However, the numbers for prior works are either missing, from a different task, or not standardized on CARLA. Multiple issues:

1. The core contribution of the authors is causal linking. The only other methods that offer "L3" level causal linking are DriveVLM CoT and DriveLM graph QA. However, they are not compared with CSN in the paper. This is a fundamental requirement. Either test CSN on their tasks, OR re-implement these methods on the proposed dataset, OR take the fundamental components of their methods and provide an apples-to-apples comparison.

2. Are the SteerVLA numbers on CARLA closed-loop or something else?

3. Are the numbers from LMDrive, TLS-Assist, and GraphPilot on the same proposed dataset? If not, please use a standardized dataset in this table, OR provide a comparison of all of the above methods on your dataset with component-wise ablations.

### **Lack of Experimental Rigour and Usefulness of Safety Supervision**

1. In Section 4.3, there needs to be a component-wise ablation for this method. Furthermore, there needs to be a clear Simplex-based baseline in Table 3. TTC safety, to my understanding, is too simple when compared to a basic Simplex-based baseline.

2. Furthermore, as per the results in Table 3, the benefits of safety supervision are not clear. Adding it simply degrades performance. Why not just use CSN? That gives the best results anyway.

3. Additionally, please provide exact experimental conditions for TTC Safety with references to prior use of these conditions from the literature.

4. φ₁ and φ₂ never trigger across the 96 route-checks. A safety guarantee that never triggers means one of two things: (1) the failure mode does not exist in the dataset, or (2) it is not tuned properly. Either way, the empirical case to justify the need for the safety supervisor is absent. Using CSN as the sole contribution of the paper and removing the safety supervisor might be a cleaner contribution of this work.

### *Minor writing issues*

1. Figure 2 is referenced before Figure 1. Please switch Figure 2 and Figure 1. that is where your contribution lies; the pipeline is not as important relatively speaking.

2. In Section 3.3, the text discusses the "39.1%" result. This is not the place to talk about it. You should simply say that you are creating a baseline here. Results can come later.

3. DS is used early on but defined later in Section 5. Move the definition to the Abstract or Introduction. Going into Section 3.3, the reader is looking for the DS definition.

### *Overclaiming safety guarantees*

On page 3, paragraph 3, the authors claim that the "Runtime safety supervisor... [provides] safety guarantees that training-time alignment cannot offer." If you want to claim a guarantee, either give a theoretical proof and show qualitative examples where this happens, or please reduce this claim. Furthermore, given that I have not seen benefits of the runtime safety supervisor on top of CSN.

**Requested Changes:**

Explained above

---

> ### Author Response · Authors · 2026-05-08
> **Response to Reviewer uZ8t**
>
> # Response to Reviewer uZ8t
>
> We thank the reviewer for the constructive feedback. In the revised PDF, **blue [R1-x]** additions respond to your comments, **orange [R2-x]** to Reviewer 2g4t, **teal [R3-x]** to Reviewer Lwcg. Changes are cited by section and page. Besides the colored, tagged changes, the revision includes minor unmarked copy-edits (wording and punctuation) that do not alter any claim.
>
> ## R1. Apples-to-apples comparison with DriveVLM and DriveLM
>
> We follow the review's third option (apples-to-apples comparison of fundamental components). On our multi-town LMDrive benchmark (16 routes, 8 towns, $N=5$, 95% bootstrap CIs), we re-implement DriveVLM's text-structural pattern and DriveLM's Graph VQA pattern at a matched token budget, holding the underlying VLA, scene state, route set, and CARLA conditions fixed. We do not retrain DriveVLM or DriveLM, because that would entangle text format with training. Each baseline has a *compressed* variant fitting LMDrive's 64-token-per-channel budget and a *verbatim* variant reproducing the paper's literal layout.
>
> | Configuration (Original LMDrive) | DS (95% CI) |
> |---|---|
> | LMDrive baseline | $32.54 \pm 3.00$ |
> | + DriveVLM-style, compressed | $31.02 \pm 4.95$ |
> | + DriveVLM-style, verbatim | $8.43 \pm 0.79$ |
> | + DriveLM-style, compressed | $28.82 \pm 3.30$ |
> | + DriveLM-style, verbatim | $5.63 \pm 0.42$ |
> | + Flat Text (ours, no causal) | $38.71 \pm 1.44$ |
> | **+ CSN (ours)** | $\mathbf{42.67 \pm 2.74}$ |
>
> **Compressed variants match the baseline; verbatim variants degrade DS by 24–27 points despite using more tokens than CSN.** Schema labels (`Object:`, `Characteristic:`, `Influence:`, `Q_n (Context:...)`) are out-of-distribution for LMDrive's frozen LLaMA, while CSN's connectives (BUT, BEFORE, BECAUSE) come from the pretraining vocabulary. The new §5.4 (pp. 16–17) reports this in Table 4 (p. 16); the contributions list (p. 2) adds it as item (3); the Table 1 caption (p. 7) separates literature-reported rows (SteerVLA, TLS-Assist, GraphPilot) from the apples-to-apples block.
>
> ## R2. Safety supervisor demoted to experimental condition
>
> **The runtime supervisor is now an experimental condition (parallel to PL-DPO-NLL), not a contribution.** Changes by section:
>
> - Title (p. 1): "Causal Scene Narration: Inference-Time Text Restructuring for Vision-Language-Action Driving".
> - Abstract (p. 1) reports the negative result on combining CSN with runtime supervision instead of headline IS/TTC numbers.
> - §1 (p. 1) removes the "no runtime safety guarantees" weakness paragraph; "three weaknesses" is now "two".
> - §2.2 (p. 4) rephrases "our semantic monitor" as "the direction-conflict monitor we evaluate as an experimental condition".
> - §4.3 (p. 9) retitled "Runtime Direction-Conflict Monitor (Experimental Condition)". §4.3.3 (p. 11) states we add no external Simplex baseline because the supervisor itself is a Simplex instance.
> - §5.1 (p. 12) question 3 rewritten as a compositional question.
> - §5.2.3 (p. 14) retitled "Comparing TTC and Direction-Conflict Monitors as Experimental Conditions"; no positive supervisor finding claimed.
> - §6 (p. 19) Finding 3 replaced with the L3 baseline comparison; Limitations, Future Work, and Ethics updated.
> - Contributions list (p. 2) adds the negative result on naive runtime supervision as item (4).
>
> The TTC threshold of 2.0 s follows Hayward (1972) and Vogel (2003) in §5.2.3 (p. 14).
>
> ## R3. Overclaiming safety guarantees
>
> Hedged in §4.3.3 (pp. 10–11): the "semantic-level safety guarantee" paragraph is replaced with **"This architecture is intended as an additional safety layer rather than a formal guarantee"**, and passive control clamping (§5.2.4, p. 14) is the only mechanism with empirical effect. The §1 sentence that made the overclaim was removed with the supervisor demotion (R2).
>
> ## R4. Minor writing issues
>
> **(a)** Figure ordering: the architecture figure moved into §4. Fig. 1 is now the causal-comparison illustration (p. 2), Fig. 2 the CSN pipeline (p. 5), Fig. 3 the architecture (p. 8).
>
> **(b)** §3.3 (p. 6): the premature "39.1%" figure is replaced by a forward reference; it first appears in §5.2.1 (p. 13).
>
> **(c)** DS is expanded on first use in the abstract (p. 1); IS no longer appears in the abstract and is first expanded in §5.1 (p. 12).
>
> ## Cross-reviewer changes
>
> For Reviewer 2g4t, §5.3 (pp. 14–15) adds paired bootstrap CIs on the structure-fraction decomposition and a bootstrap on the cross-configuration asymmetry (not distinguishable from zero at $N=5$). For Reviewer Lwcg, CSN pseudocode appears as Algorithm 1 in §4.2.2 (p. 9); the §5.2.2 title (p. 14) limits the robustness claim to the two tested LoRA weight configurations; a residual sim-to-real gap is named in Limitations (§6, p. 19).

---

### Review · Reviewer_2g4t · 2026-06-08

**Summary Of Contributions:**

The paper introduces Causal Scene Narration (CSN), a  text-restructuring method for Vision-Language-Action (VLA) autonomous driving. The core hypothesis is that what helps VLA models is not the quantity of textual scene information but its causal structure—specifically, explicitly linking navigation intent to relevant environmental constraints via linguistic connectives (BUT/YIELD_BEFORE/BECAUSE). The contributions are:

- CSN pipeline: A  text-enrichment method built on three principles (quantitative grounding, structured separation, intent-constraint causal alignment).
- Runtime safety supervisor: A Simplex-architecture monitor enforcing semantic safety properties (direction consistency φ₁, stuck-detection φ₂) in Signal Temporal Logic, contrasted against a reactive Time-To-Collision baseline.
- Multi-town CARLA evaluation: A ten-configuration ablation over 16 routes / 8 towns / N=5, with a perception-noise robustness study and a notable negative result that CSN+Safety degrades performance due to passive control clamping.

**Audience:**

Yes

**Audience Explanation:**

The information-vs-structure decomposition is a clean, transferable idea relevant well beyond driving—it speaks to a general question about why structured/CoT-style text helps LLMs (attention scaffolding vs. added information)

**Broader Impact Concerns:**

No conc1ern.

**Claims And Evidence:**

No

**Claims Explanation:**

- Looking at Table 2., the central paper claim that causal structure contributes independently of information content does  not appear fully support. Flat-text already improve significantly over the LMDrive baseline, suggesting that information content is important, althought the CSN approach is still slightly better. Flat-text and CSN, have very close DSDrive, however, in the PL-DPO-NLL. Abstract and Introduction should reflect better those results.

- IThe perception-noise result is genuinely puzzling if I understand them correctly. Every noise condition yields higher mean DS than the clean baseline (45.1–46.0 vs. 42.7). The authors attribute this to N=5 sampling variance, but this raise concern about the robustness of the observation.

**Requested Changes:**

1. Update abstract and intro to reflect the more nuance picture depicted in table 2. In particular, CSN provide a limited improvement over the flat-text baseline which should lead to more nuance claim.
2. Explain the noise-addition results. Why does each noise-injection regime lead to better results than the clean baseline? Maybe this suggest that N need to be increased to get more reliable observation in general in the paper.
3. PL-DPO-NLL is presented as a contribution, yet it's unclear from table 2. that it provides an improvement over the LLMDrive baseline. It would be nice to describe what is the advantage of PL-DPO-NLL training.

---

> ### Author Response · Authors · 2026-07-05
> **Point-by-point response: bootstrap CIs added throughout, noise paradox addressed with sign test + falsification control run during discussion, PL-DPO-NLL reframed. Changes colored orange [R2-x] in revised PDF.**
>
> # Response to Reviewer 2g4t
>
> We thank the reviewer. Colored additions in the revised PDF are keyed: **orange [R2-x]** to your comments; **blue [R1-x]** / **teal [R3-x]** to Reviewers uZ8t / Lwcg. Unmarked minor copy-edits do not alter any claim. (The table the review calls Table 2 is Table 3 in both submission and revision.)
>
> ## Concern 1. Information-vs-structure decomposition is overclaimed
>
> The reviewer's reading is right: Flat Text alone recovers most of the gain (+6.17 of +10.13 DS on Original LMDrive), and on PL-DPO-NLL CSN beats Flat Text by only +1.08 DS (CI crosses zero).
>
> **Abstract and Introduction updated as requested.** The Abstract (p. 1) now leads with "information content carries most of CSN's gain on both weight configurations" and gives both structure fractions with CIs; the Introduction and Contribution 2 (p. 2) say the causal-phrasing share is measurable on Original LMDrive but not separable from zero on PL-DPO-NLL, with the gap between the shares itself within noise.
>
> To quantify this, the revision adds a paired repetition-level (rep-level) bootstrap (10,000 resamples; same index across baseline / Flat Text / CSN; baseline bootstrapped, not held fixed) and a per-route paired bootstrap (§5.3, pp. 14–15):
>
> | | Structure fraction (95% CI) | P(>0) | Wins/ties/losses | Per-route diff, DS (95% CI) |
> |---|---|---|---|---|
> | Original LMDrive | 39.1% [6.6, 71.4] | 0.986 | 5/4/7 | +3.96 [-3.86, +13.41] |
> | PL-DPO-NLL | 13.5% [-244, +41.9] | 0.664 | 5/6/5 | +1.08 [-4.44, +6.99] |
> | Asymmetry (difference) | +25.6% [-23.8, +306.4] | 0.837 | — | — |
>
> P(>0) is the bootstrap probability that the column-2 quantity is positive. The asymmetry CI includes zero, so the cross-configuration asymmetry is suggestive rather than established. Wins/ties/losses count routes (of 16) where CSN beats Flat Text by more than 0.5 DS; the 5/4/7 split shows the mean advantage comes from a few large wins, not uniform improvement. The same intervals also appear in §5.2.1 (p. 13), §5.5.4 (p. 18), and the Conclusion (p. 19).
>
> ## Concern 2. Noise-ablation paradox
>
> The reviewer is right that "attributable to N=5 sampling variance" read as dismissive. The revision responds as follows (§5.5.2, pp. 17–18; Table 5, p. 17):
>
> - **Sign test with seed disclosure.** Under independence and a symmetric null, P(all four noisy means exceed clean) ≤ 1/16. Table 5's caption discloses shared seeds (noise RNG 42; default traffic-manager), so the true joint probability is somewhat larger; the pattern still needs a mechanism-level explanation.
> - **Three falsifiable mechanisms** (p. 17). We hypothesized three sources for the lift: (i) text-manifold regularization (falsifier: effect vanishes if noise applies only to the textual rendering); (ii) asymmetric-cost buffer, since CSN encodes small reported gaps as earlier YIELD_BEFORE clauses (falsifier: reverses under negative-only noise); (iii) degenerate-tie breaking in LLM decoding (falsifier: lift appears for any dynamic text, e.g., Flat Text + noise).
> - **New control experiment (N=5).** To test these mechanisms during the discussion period, we ran Flat Text + Severe noise. It scores 37.66 ± 2.36 vs 38.71 ± 1.88 clean (rep-level SD; Table 3's ±1.44 is a bootstrap CI); diff -1.05 DS, CI [-3.35, +1.38] (added to Table 5 as a control row).
> - **Implication.** The lack of lift on Flat Text disfavors the text-generic explanations (i) and (iii), leaving the CSN-specific mechanism (ii) as the most consistent, though this disfavors alternatives rather than confirming (ii).
> - A further Clean repetition gives N=6: 42.3 ± 3.5, CI [39.5, 45.0] (Table 5). Noise rows remain N=5 with CIs compatible with Clean; the paradox persists at the higher N.
> - We attempted N=10, but compute-time constraints within the discussion window limited us to N=6 Clean and N=5 noise (Limitations, §6, pp. 19–20); a two-sample test would not be powered until N ≥ 20, left to follow-up work.
>
> The injected noise is zero-mean symmetric Gaussian clipped at 1 m, and we make no claim of a saturating response (Severe vs Extreme is within bootstrap noise).
>
> ## Concern 3. PL-DPO-NLL does not improve over the LMDrive baseline
>
> Agreed (Reviewer uZ8t flagged the same). PL-DPO-NLL is removed from the §1 contributions and labeled an "experimental condition" in §1 (p. 2), the §4.4 title (p. 11), §5.1 (p. 12), and the §5.2.1 title (p. 13). We retain it because the §5.3 decomposition needs a second weight configuration. The asymmetry CI includes zero, so the cross-configuration pattern is no longer presented as a finding; the Conclusion (p. 19) now describes the pattern as consistent with, but not establishing, preference learning partially internalizing causal reasoning.
>
> We will revise further if any point remains unaddressed.

---

### Review · Reviewer_Lwcg · 2026-06-22

**Summary Of Contributions:**

This paper introduces Causal Scene Narration (CSN), an inference-time method that restructures the text inputs of Vision-Language-Action (VLA) models for autonomous driving so that the causal relationship between navigation intent and environmental constraints is made explicit. The paper further proposes a Simplex-style Runtime Safety Supervisor that monitors VLA waypoints against a semantic safety envelope. Experimental results on LMDrive architectures demonstrate performance gains in a multi-town CARLA evaluation.

**Strengths**:
1. The idea of reconstructing the text input of autonomous driving VLAs to strengthen the causal relationship between intent and constraints is well-motivated.
2. Empirical results show that the proposed CSN method is beneficial, and that it requires neither retraining nor additional GPU cost at test time.
3. The discussion on "Why does causal structure help" is useful.

**Weaknesses**:
1. The main experiments are conducted with privileged simulation data (as discussed in Sec. 5.4.2). Analysis with Gaussian noise is provided, but there is still a gap towards real-world deployment, which is worth noting.
2. Technical details of how CSN pre-selects constraints are not clearly specified.
3. The intent-constraint causal alignment relies on a pre-defined list of three conflict types (blocking, temporal and explanatory). This design seems to introduce huge inductive bias, which warrants discussion of its expandability and coverage.
4. The statement of "CSN Robustness Across Configurations" (Sec. 5.2.2) is too strong given that only two configurations within the same architecture are tested.

**Audience:**

Yes

**Audience Explanation:**

The findings that the improved causality between the intent and constraints in the text inputs of VLAs for autonomous driving will interest the audience in the autonomous driving field and VLA field.

**Broader Impact Concerns:**

No additional Broader Impact Statement is required.

**Claims And Evidence:**

Yes

**Claims Explanation:**

The submission's claim that CSN is effective is supported by empirical results from a multi-town closed-loop CARLA evaluation. The ablations show that the improvement stems from better causal structure rather than from additional information alone.

**Requested Changes:**

**Major points**:
1. Specify the constraint filtering and ranking procedure in CSN (the relevance filter, the urgency-ranking function, and how many constraints are kept), ideally with pseudocode, so the method is reproducible.
2. Modify "CSN Robustness Across Configurations" (Sec. 5.2.2) and related claims: only two weight configurations of a single architecture (LMDrive/LLaMA-7B) are tested, which the paper itself concedes in Sec. 5.4.3 and the Limitations.

**Minor points**:
1. Fix the cross-reference in Sec. 4.2.1: "Replacing this with a vision-based perception stack is discussed in Sec. 6" points to a one-line future-work item. The substantive treatment of perception realism is the noise ablation in Sec. 5.4.2, so the pointer should go there, and Sec. 6 should be described as future work rather than a discussion.
2. Define the L0-L3 taxonomy before it is used. Figure 2 on Page 5 is labeled with L0/L3, but the taxonomy is only defined in Sec. 3.4 on Page 5-7 either move the definition earlier or annotate the figure.

---

> ### Author Response · Authors · 2026-07-05
> **Point-by-point response: CSN pseudocode added (Algorithm 1), robustness claim scoped to the two tested configurations, cross-reference and taxonomy ordering fixed. Changes colored teal [R3-x] in revised PDF.**
>
> # Response to Reviewer Lwcg
>
> We thank Reviewer Lwcg for the careful reading and the four concrete Requested Changes. The revision addresses each Major and Minor point. In the revised PDF, edits are marked as colored additions keyed to reviewers: **teal [R3-x]** responds to your comments; **blue [R1-x]** and **orange [R2-x]** respond to Reviewers uZ8t and 2g4t. Besides the colored, tagged changes, the revision includes minor unmarked copy-edits (wording and punctuation) that do not alter any claim. (Numbering note: the noise ablation the review calls Sec. 5.4.2 is Sec. 5.5.2, p. 17, in the revision, because a new Sec. 5.4 was inserted for Reviewer uZ8t.)
>
> ## Major 1. CSN constraint filtering and ranking pseudocode
>
> Algorithm 1 (CSN constraint filtering and ranking) is added to §4.2.2 (p. 9); it formalizes the two-stage filter-then-rank pipeline. The surrounding paragraph specifies every numeric constant: the relevance threshold (τ = 0), pre-filter caps (at most 3 vehicles, 2 pedestrians), the three relevance signals (distance decay, conflict-zone multiplier, motion modifier), the classification priority blocking > temporal > explanatory, and the token-budget bound that determines K, the number of constraints kept per frame (T_max = 58 tokens per channel, typically K ≤ 5). The pseudocode matches the implementation used in all experiments.
>
> ## Major 2. Modified robustness-across-configurations claim
>
> The original "CSN Robustness Across Configurations" overreached given only two weight configurations of a single architecture. The §5.2.2 title (p. 14) now reads "CSN Benefits Hold Across the Two LoRA Weight Configurations Tested", the body names the two configurations (the original release weights and our PL-DPO-NLL LoRA variant), and it adds the requested disclaimer: *"We do not claim robustness across architectures, model scales, or training recipes beyond these two configurations."* The same softening appears in the Abstract (p. 1), Introduction Contribution 2 (p. 2), the Conclusion first finding (p. 19), and Limitations item (2) (p. 19).
>
> ## Minor 1. §4.2.1 cross-reference
>
> The §4.2.1 pointer (p. 8) no longer reads "discussed in Sec. 6"; it now points to the perception noise ablation, §5.5.2 (pp. 17–18; formerly Sec. 5.4.2). §6 is now described as a future-work item rather than a discussion of perception realism.
>
> ## Minor 2. L0–L3 taxonomy ordering
>
> We retained the full L0–L3 definition in §3.4 (p. 7) but added a one-sentence definition where the labels first appear, at the beginning of §3.1 (p. 4), covering L0 (isolated commands), L1 (structured facts), L2 (entity-level scene graphs), and L3 (explicit intent-constraint causal dependence). This makes Figure 1's L0/L3 markers (p. 2) readable without forcing the reader forward to §3.4.
>
> ## Weakness 1. Privileged simulation data and sim-to-real gap
>
> Limitations item (1) in the Conclusion (p. 19) now states the gap explicitly: all CSN inputs derive from CARLA's privileged API, so a residual sim-to-real gap remains; the noise ablation is the only sim-to-real evidence we offer and does not substitute for on-vehicle validation.
>
> ## Weakness 2. CSN constraint pre-selection details
>
> Addressed by Algorithm 1 under Major 1 (p. 9).
>
> ## Weakness 3. Three conflict types as inductive bias
>
> We added a defense paragraph at the end of §3.2 (p. 6) addressing the coverage and expandability of this design:
>
> - (i) Coverage of observed failures: each of the four observed failure causes on `benchmark_short` maps to one of the three conflict types, so the three types cover the failure modes the benchmark exposes.
> - (ii) Extensibility without retraining: the γ function family in Eq. 2 is extensible; additional connectives (such as UNLESS, WHILE, UNTIL) can be added without retraining, since they affect only the inference-time text renderer.
> - (iii) Explicit limitation: the three-connective restriction remains untested for non-driving scene types. Limitations item (4) in the Conclusion (p. 19) records this caveat.
>
> ## Weakness 4. §5.2.2 claim too strong
>
> Addressed under Major 2 (p. 14).
>
> ## Cross-reviewer changes
>
> Changes for the other reviewers touch overlapping sections. For Reviewer 2g4t, who requested bootstrap 95% CIs (Table 3 already carried them in the submission), the new material is CIs on the structure fractions (§5.2–5.3, pp. 13–15), a bootstrap test on the cross-configuration asymmetry in structure fractions (95% CI [-23.8%, +306.4%], including zero), and a Flat Text noise-control row in Table 5 (p. 17). Reviewer uZ8t requested demotion of the runtime supervisor from a contribution to an experimental condition, now reflected in the Abstract, Introduction, §4.3 and §4.4 headers, and §5.2.3 (pp. 1–2, 9, 11, 14). These changes alter no empirical results or qualitative conclusions on CSN.